# On the Generalization Gap in Self-Evolving Language Model Reasoning

**Zhenting Qi** [1,2]  **Susanna Maria Baby** [3]  **Stefanie Anna Baby** [3]  **Kan Yuan** [3]  **Andrew Tomkins** [2]  **Tu Vu** [4,2]  **Da-Cheng Juan** [2]  **Cyrus Rashtchian** [2]

## Abstract

Recent work suggests that large language models (LLMs) can improve through *self-evolution* (SE), using supervision signals generated by the model itself. In this work, we ask: under a strict closed-loop setup, where the SE algorithm has access only to an unlabeled prompt set and a base model, how close can internally generated supervision come to oracle-supervised training? We analyze four representative strategies in a unified offline self-evolution framework, including single-round verification, multi-turn revision with feedback, iterative training, and curriculum learning. Our primary experiments use Knights and Knaves (KK) logical reasoning tasks, which provide deterministic solutions, controlled difficulty levels, and a clean testbed for easy-to-hard generalization. We first show that SE consistently improves over the base model, but plateaus after excessive training compute is invested, and eventually still leaves a non-trivial gap to oracle supervision. We find that multi-turn critic-revision with large models could reach strong self-evolution performance, where Gemma 12B nearly matches oracle-supervised training. Beyond KK, we also evaluate SE on real-world reasoning benchmarks, where gains are also modest. Overall, our results characterize when closed-loop SE can help, and show how internally generated supervision remains insufficient under this minimal formulation.

## 1. Introduction

Large language models (LLMs) have made strong progress on complex reasoning tasks (Comanici et al., 2025; Yang et al., 2025; DeepSeek AI, 2025). A central driver of this progress has been post-training techniques that refine model

outputs using feedback signals. Paradigms such as Supervised Finetuning (SFT), Reinforcement Learning from Human Feedback (RLHF), and Reinforcement Learning from Verifiable Rewards (RLVR) (Lambert et al., 2024; Gao et al., 2024; Wang et al., 2025; Shen et al., 2025) have become standard tools for improving reasoning. However, these approaches typically require human-labeled preference data, verifiable rewards, or ground-truth solutions. Recently, an alternate direction has attracted increasing attention: *can models improve using supervision signals generated only by the model itself?* We refer to this family of approaches as *self-evolution* (SE), and focus in particular on the *closed-loop* setting where all supervision is produced internally by the model being improved. In closed-loop SE, a model may improve by verifying its own solutions (Shafayat et al., 2025; Zeng et al., 2025; Zhang et al., 2025c; Piterbarg et al., 2025; Zhao et al., 2025a) or by generating and exploiting structured forms of internal feedback (Huang et al., 2025b; Zuo et al., 2025). More generally, methods such as INTU-ITOR (Zhao et al., 2025b), LSP (Kuba et al., 2025), and EMPO (Zhang et al., 2025a) primarily employ online RL with intrinsic, model-based rewards to enable learning.

Despite promising empirical results, SE exists in tension with a growing body of evidence highlighting the limitations of learning from synthetic or self-generated data. Prior studies have documented failure modes such as model collapse (Dey & Donoho, 2024; Shumailov et al., 2024; Wenger, 2024), theoretical barriers to learning from synthetic distributions (Amin et al., 2026), and diminishing returns from reinforcement learning without verifiable rewards (Setlur et al., 2025; Song et al., 2024). Moreover, Zhang et al. (2025c) demonstrate that simple generation–verification pipelines, such as continued pre-training on rejection-sampled data, are insufficient to yield sustained self-improvement. Taken together, these results motivate a focused empirical question: *How close can closed-loop SE come to oracle-supervised training?* Importantly, this question is not the same as asking for the ultimate ceiling of all possible self-evolution methods. Self-evolution can be instantiated in many ways, including settings with external tools, symbolic environments, multiple models, stronger verifiers, online interaction, or other sources of additional structure. Our goal is instead to study a deliberately minimal

---

[1]Harvard University [2]Google Research [3]Google [4]Virginia Tech. Correspondence to: Cyrus Rashtchian <cyroid@google.com>, Zhenting Qi <zhentingqi@g.harvard.edu>.

*Proceedings of the 43$^{rd}$ International Conference on Machine Learning*, Seoul, South Korea. PMLR 306, 2026. Copyright 2026 by the author(s).

and controlled formulation, where the learner is given only an unlabeled prompt set $\mathcal{D}$ and a base model $\mathcal{M}$. Beyond $\mathcal{D}$, all reasoning traces, solutions, rewards, feedback, and preference labels must be generated from $\mathcal{M}$ itself. This formulation isolates the gap that can be closed by internal model-generated signals alone, without external supervision or a verifiable environment.

Under this closed-loop formulation, many recently proposed SE approaches can be viewed as different instantiations of a common generator–verifier framework, differing in how internal signals are extracted, reused, and structured. Rather than proposing a new training algorithm, our goal is to systematically analyze how close increasingly structured and computationally intensive SE strategies can approach oracle-supervised preference optimization. Our primary analysis is conducted in an intentionally simple yet clean setting using the *Knights-and-Knaves* (KK) logical reasoning tasks (Xie et al., 2024). Each task instance describes a group of people who are either *knights* (always truthful) or *knaves* (always lying). The task is to infer each person's identity from their statements, such as claims about themselves or other people. Because KK is deterministically constructed, it provides exact solutions that are uniquely verifiable. It also mitigates data contamination concerns, as the models we evaluate still struggle to achieve high accuracy. Problem difficulty is parameterized by the number of people, allowing us to vary complexity in a controlled manner. By training on simpler instances and evaluating on harder ones, we can measure easy-to-hard generalization and study curriculum learning in a principled setting.

We empirically show that closed-loop SE techniques consistently improve over the base model, but often remain substantially below oracle-supervised training. For example, on the Knights-and-Knaves benchmark, Gemma 3 4B achieves an average accuracy of 31.0%. Training with oracle preference signals raises accuracy to 53.3%. In contrast, closed-loop SE improves accuracy to **40.7%** with single-round verification, **42.2%** with multi-turn feedback, **44.1%** with iterative training, and **44.8%** with curriculum learning. Although increasingly structured SE yields monotonic gains in this setting, *a persistent performance gap of around 8–13% remains relative to the oracle setting.*

To assess the robustness of these findings beyond logical reasoning, we additionally study SE on OpenThoughts3 (Guha et al., 2025) reasoning tasks and evaluate the resulting models on other downstream reasoning benchmarks, and show that offline preference learning performs on par with more expensive online methods, but also show that the improvement is modest and might even degrade sometimes. Accordingly, our objective is not to maximize benchmark performance or to isolate every source of error for SE, but to contextualize the self-improvement magnitude and limitations by comparing a simple and general framework against stronger and more resource-intensive training paradigms.

In summary, we make the following contributions:

- **Systematic analysis of closed-loop SE:** We provide a unified comparison of SE methods under a strict closed-loop framework, isolating their effects relative to oracle-supervised preference optimization. We evaluate multiple methods on the Knights-and-Knaves benchmark, which avoids data contamination while enabling a controlled study of training, verification, revision, iteration, and curriculum strategies.

- **Effect of internal supervision quality:** To maximize the improvement from SE, it is crucial to extract reliable verifier signals from the model itself. For example, we find that multi-turn feedback leads to the best performance, especially for a larger 12B model, achieving 52.8% accuracy and nearly matching the oracle-based 53.6%. However, constructing supervision in this way incurs a high computational cost.

- **Bootstrapping and generalization:** In addition to studying overall performance, we investigate easy-to-hard transfer. We find that iterative training and curriculum learning can enhance the model's ability to solve harder examples, such as KK instances with 6–8 people. Thus, closed-loop SE can produce consistent gains, but its ability to improve out-of-distribution generalization remains limited.

- **Persistent gap to oracle supervision under the closed-loop constraint:** Even with increasingly structured and computationally intensive SE strategies, performance often remains below oracle-supervised training. We observe this pattern on both the controlled Knights-and-Knaves data and the real-world OpenThoughts3 setting. Overall, our results suggest that, under a strict closed-loop formulation, internal feedback alone is not a general substitute for verifiable rewards or curated ground-truth solutions. SE may help surface a model's latent abilities, but in our experiments it does not reliably teach the model to solve much harder problems without external signal.

**Conflict of Interest Disclosure.** The authors are currently or previously employed by Google, which leads the development of the Gemma family of models, which is among the ones evaluated in this paper.

## 2. Preliminaries

There are many ways to use a model to impute a ground truth signal, and we aim to study a representative spread of natural options. In particular, we frame these options as increasingly complex preference data collection, using only

## Single-Round Self-Evolution

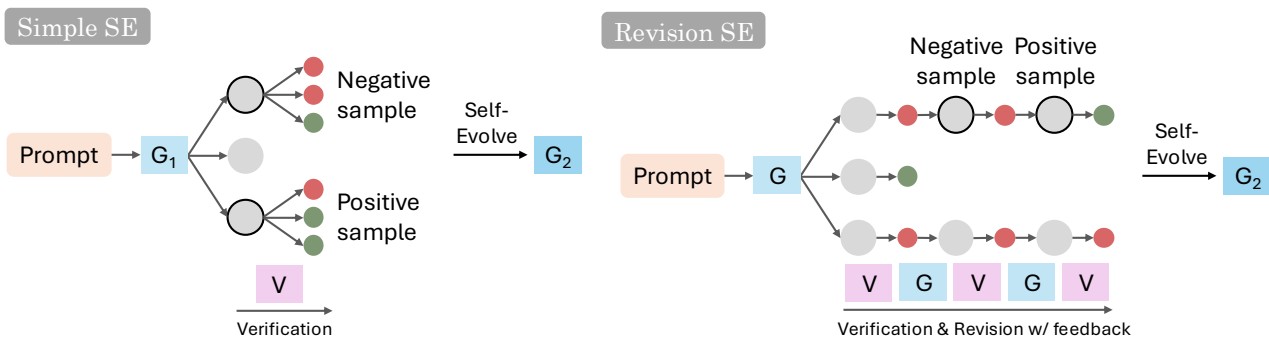

## Multi-Round Self-Evolution

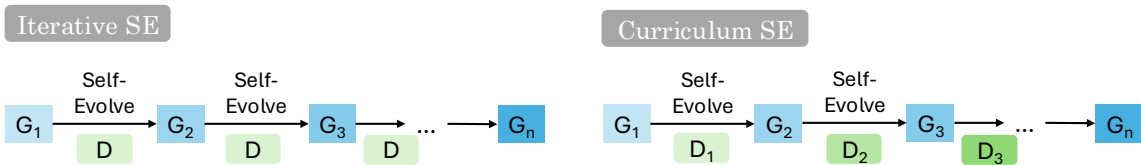

*Figure 1.* **Taxonomy of self-evolution strategies analyzed in this paper** (from simple to complex). The top row illustrates *single-round self-evolution*, where preference data are constructed from a single generator–verifier interaction, including (left) direct verification and (right) solution enhancement with revision and feedback. The bottom row illustrates *multi-round self-evolution*, where the generator–verifier procedure is applied repeatedly across training rounds. Within multi-round self-evolution, we distinguish iterative self-evolution, which repeatedly applies self-evolution on the same dataset $D$, from curriculum self-evolution, which trains over datasets $\{D_n\}$ ordered from easy to hard for the model to solve. Here, $D_n$ denotes the $n$-th stage in the curriculum.

feedback mechanisms from a single model. Our primary goal is to systematically study self-evolution, incorporating variations that are absent from prior work (Song et al., 2024; Sun et al., 2025b). Given the vast amount of research in this area, we defer the full related work to Section A.

At a high level, we instantiate a single base model in two roles: a *generator*, which proposes candidate solutions, and a *verifier*, which evaluates their quality. In the simplest *single-turn self-evolution* setting, the verifier forms preference pairs $(y_{\text{win}}, y_{\text{lose}})$ by labeling candidate responses. Since we cannot always expect the verifier to be better than the generator, we also explore thresholded voting to aggregate multiple verifier responses. We query the verifier multiple times per candidate. Define the *correctness rate* as the fraction of times the verifier says a response is correct. We label a response as *positive* if its correctness rate exceeds a threshold $\tau$, and as *negative* if its correctness rate is less than $(1 - \tau)$, and we discard it otherwise. This filters out ambiguous cases and yields "confident" preference pairs, extracting a reliable signal from imperfect self-assessment. In a richer *multi-turn game*, the verifier iteratively provides feedback and the generator revises its outputs, producing higher-quality alternatives. An interesting aspect of revision is that we can hope that the data improves to a point that the model can learn from it, even when it is not fully correct. Finally, we explore extensions of these variants, where we use iterative training or curriculum learning.

**Generator–Verifier Game.** A single base model $\mathcal{M}$ is instantiated in two roles using different system prompts: a *generator* $\mathcal{G}$ and a *verifier* $\mathcal{V}$.[1] Given an unlabeled prompt set $\mathcal{D}$ and a base model $\mathcal{M}$, define a generator–verifier game

$$\text{GV}(\mathcal{M}, \mathcal{D}, T) \rightarrow \mathcal{P},$$

which uses $\mathcal{M}$ as generator $\mathcal{G}$ and verifier $\mathcal{V}$, and runs for $T$ rounds. For a query $q \in \mathcal{D}$, the generator produces $k$ candidates $\hat{Y}(q) = \{\hat{y}_1, \ldots, \hat{y}_k\}$ where $\hat{y}_i \sim \mathcal{G}(\cdot \mid q)$, while the verifier assigns binary judgments $\mathcal{V}(q, \hat{y}_i) \in \{\texttt{Correct}, \texttt{Incorrect}\}$. Then, from these interactions we extract preference pairs $(y_w, y_l) \in \mathcal{P}$ if and only if $\mathcal{V}(q, y_w) = \texttt{Correct}$ and $\mathcal{V}(q, y_l) = \texttt{Incorrect}$.

**Single-turn Verification vs. Multi-turn Feedback.** In the *single-turn* case, $T = 1$ and preference pairs are obtained directly from static judgments. In the *multi-turn* case ($T > 1$), the generator refines its outputs based on verifier feedback:

$$\hat{y}^{(t+1)} \sim \mathcal{G}(\cdot \mid q, f(\mathcal{V}(q, \hat{y}^{(t)}))),$$

where $f : \{\texttt{Correct}, \texttt{Incorrect}\} \rightarrow \mathcal{X}_{\text{feedback}}$ maps verifier judgments into textual feedback prompts. A pair is extracted whenever

$$\mathcal{V}(q, \hat{y}^{(t)}) = \texttt{Incorrect}, \quad \mathcal{V}(q, \hat{y}^{(t+1)}) = \texttt{Correct}.$$

---

[1]We provide the exact prompts in Section D.

**Preference Learning.** The generator–verifier game yields a dataset of preference triples $\mathcal{D}_{\text{pref}} = \{(x, y_w, y_l)\}$, where $x$ is a prompt, $y_w$ is a preferred response, and $y_l$ is a dispreferred one. Preference learning fine-tunes a policy $\pi_\theta$ so that preferred responses are assigned higher probability than dispreferred ones. We apply *Direct Preference Optimization* (DPO) (Rafailov et al., 2023) which refines $\pi_\theta$ relative to a fixed reference policy $\pi_{\text{ref}}$ by minimizing

$$\mathcal{L}_{\text{DPO}}(\pi_\theta; \pi_{\text{ref}}) = -\mathbb{E}_{(x, y_w, y_l) \sim \mathcal{D}_{\text{pref}}} \Big[ \log \sigma(\beta \log \frac{\pi_\theta(y_w|x)}{\pi_{\text{ref}}(y_w|x)} \\ - \beta \log \frac{\pi_\theta(y_l|x)}{\pi_{\text{ref}}(y_l|x)}) \Big]$$

where $\beta > 0$ is a parameter controlling the sharpness of preference alignment. Intuitively, this loss increases the relative likelihood of $y_w$ over $y_l$ while keeping $\pi_\theta$ close to reference policy $\pi_{\text{ref}}$, ensuring both preference alignment and stability during fine-tuning. Unlike other self-training DPO approaches (Wang et al., 2024a), one model generates reasoning traces *and* uses internal signals to impute labels.

**Iterative Preference Learning.** We may apply GV repeatedly. Starting from $\mathcal{M}_0 = \mathcal{M}$, define

$$\mathcal{P}_t = \mathsf{GV}(\mathcal{M}_{t-1}, \mathcal{D}_t, T), \mathcal{M}_t = \texttt{Finetune}(\mathcal{M}_{t-1}, \mathcal{P}_t).$$

This yields a sequence $\{\mathcal{M}_t\}_{t=1}^T$ that progressively refines reasoning ability. Unlike online RL, all updates are offline since $\mathcal{P}_t$ is fixed once generated. This approach mimics supervised iterative DPO (Pang et al., 2024).

**Curriculum Learning.** If prompts can be partitioned by difficulty, $\mathcal{D} = \mathcal{D}_{\text{easy}} \cup \mathcal{D}_{\text{hard}}$, we first generate $\mathcal{P}_{\text{easy}} = \mathsf{GV}(\mathcal{M}, \mathcal{D}_{\text{easy}}, T)$ and fine-tune on it, before proceeding to $\mathcal{P}_{\text{hard}}$. While curriculum learning is a standard approach in supervised model training, its influence on self-evolution preference optimization has been less well studied.

**Oracle Supervision.** For all of the above variations, we also consider an *oracle verifier* setting. Here, we use the ground-truth solutions along with a deterministic function (exact-match) to label examples as correct or incorrect. In our experiments, we will incorporate one or more rounds of oracle-labeled solutions into our self-evolution strategies.

### 2.1. Experiment Setup

**Models.** We use the `gemma-3-it` (Gemma Team, 2025) and `Qwen-2.5-Instruct` (Yang et al., 2025) families as base models. Since the same model is instantiated as both generator and verifier, we employ instruction-tuned variants rather than raw base models.

**Datasets.** Our controlled experiments use the *Knights-and-Knaves* (KK) (Xie et al., 2024) dataset for reasoning. Each instance describes a group of inhabitants who are either *knights* (always truthful) or *knaves* (always lying). The task

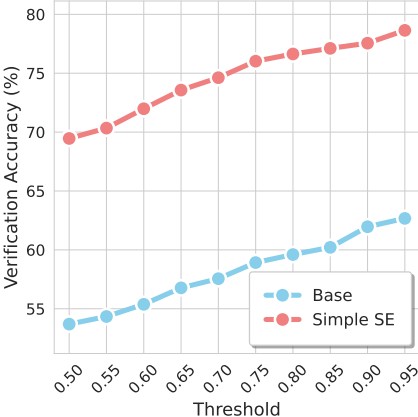

*Figure 2.* Verifier accuracy on KK train set for `gemma-3-4b-it` and its SimpleSE variant under many thresholds.

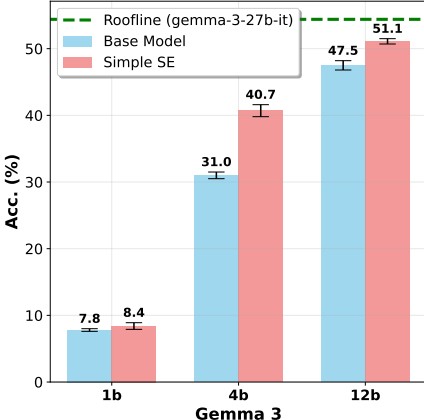

*Figure 3.* Effect of model size on SimpleSE accuracy. We train models on KK instances w/ 2–3 people, evaluate w/ 2–8 people.

is to infer each inhabitant's identity from their statements. The difficulty scales with the number of people, as the search space grows exponentially and demands deeper inference. This structured setting provides a testbed for isolating the effects of self-evolution. Unsupervised training only uses unlabeled prompts, without ground-truth solutions.

**Evaluation Protocol.** We report exact-match accuracy, requiring the model's output to perfectly match the ground-truth solution. This metric eliminates ambiguity from partial matches. For each query, we generate one sample using temperature 0.7 and average results over four random seeds.

## 3. Spectrum of SimpleSE Variants

We begin with the simplest setting. A single model serves as generator and verifier, and the verifier directly judges the quality of responses without iterative feedback. This approach has been well-studied before, but we include it as a baseline to build off for other self-evolution methods.

*Table 1.* **Iterative Training Results on KK.** Accuracy (%) is averaged over subsets (2–3, 4–5, and 6–8 people) with std. dev. in parentheses. Oracle results (in ForestGreen) use ground-truth verification. Other rows compare SimpleSE thresholds $\tau$. Three rounds of unsupervised DPO improve accuracy from 31.0% to 44.1%, approaching the 46.6% achieved with a single oracle round. However, adding a final oracle-verification round boosts accuracy to 53.2% or 52.6%, highlighting the persistent generalization gap of SimpleSE.

| Model/Algorithm | 2–3 ppl. | 4–5 ppl. | 6–8 ppl. | All |
|---|---|---|---|---|
| gemma-3-4b-it (baseline) | 62.0 (1.7) | 31.0 (0.9) | 10.3 (1.3) | 31.0 (1.3) |
| SimpleSE, $\tau = 0.6$ | 70.9 (1.9) | 45.4 (3.8) | 17.5 (2.9) | 40.7 (2.8) |
| Oracle Verifier | 78.4 (1.8) | **52.6** (2.1) | 21.4 (1.4) | 46.6 (1.7) |
| SimpleSE, $\tau = 0.6 \rightarrow$ SimpleSE, $\tau = 0.5$ | 69.5 (1.9) | 44.8 (2.7) | 18.1 (1.3) | 40.4 (1.9) |
| $\rightarrow$ SimpleSE, $\tau = 0.6$ | 74.2 (2.1) | 46.9 (2.5) | 20.3 (0.8) | **43.3** (1.6) |
| $\rightarrow$ SimpleSE, $\tau = 0.7$ | 71.5 (1.8) | 46.8 (1.9) | **20.8** (2.0) | 42.7 (1.9) |
| $\rightarrow$ SimpleSE, $\tau = 0.8$ | 72.2 (1.6) | 48.1 (2.2) | 18.6 (1.3) | 42.4 (1.7) |
| $\rightarrow$ Oracle Verifier | 82.4 (0.8) | 58.6 (2.3) | 30.2 (2.5) | **53.2** (1.9) |
| SimpleSE, $\tau = 0.6 \rightarrow$ SimpleSE, $\tau = 0.6 \rightarrow$ SimpleSE, $\tau = 0.5$ | **75.2** (1.6) | **49.6** (2.0) | 19.7 (2.0) | 44.1 (1.9) |
| $\rightarrow$ SimpleSE, $\tau = 0.6$ | 74.5 (1.4) | 46.0 (1.8) | 18.8 (1.7) | 42.5 (1.7) |
| $\rightarrow$ SimpleSE, $\tau = 0.7$ | 70.8 (1.6) | 46.3 (2.4) | 16.3 (1.1) | 40.4 (1.6) |
| $\rightarrow$ SimpleSE, $\tau = 0.8$ | 72.2 (2.6) | 45.9 (2.4) | 20.7 (1.7) | 42.6 (2.2) |
| $\rightarrow$ Oracle Verifier | **85.0** (1.1) | **61.9** (1.8) | 25.0 (2.4) | 52.6 (1.9) |

*Table 2.* **Curriculum Learning Results on KK.** Accuracy (%) is averaged over subsets (2–3, 4–5, and 6–8 people) with std. dev. in parentheses. Rows compare SimpleSE thresholds $\tau$. Oracle results (in ForestGreen) use ground-truth verification. Similar to the iterative DPO results, curriculum learning performs better than random mixing. However, using the oracle verifier leads to substantially better results in all cases, with roughly a 5% gap above the best SimpleSE setting.

| Model/Algorithm | 2–3 ppl. | 4–5 ppl. | 6–8 ppl. | All |
|---|---|---|---|---|
| gemma-3-4b-it (baseline) | 62.0 (1.7) | 31.1 (0.9) | 10.3 (1.3) | 31.0 (1.3) |
| KK2345 w/ SimpleSE, $\tau = 0.5$ | 68.6 (1.7) | 44.3 (1.6) | 17.6 (1.5) | 39.8 (1.6) |
| w/ SimpleSE, $\tau = 0.6$ | 67.2 (1.5) | 39.9 (2.1) | 14.7 (1.5) | 36.9 (1.6) |
| w/ SimpleSE, $\tau = 0.7$ | 71.0 (1.8) | 42.9 (1.1) | 15.4 (1.1) | 39.1 (1.3) |
| w/ SimpleSE, $\tau = 0.8$ | 72.9 (2.5) | 46.1 (1.9) | 16.7 (1.8) | **41.1** (2.0) |
| w/ Oracle Verifier | 80.9 (1.6) | 54.4 (1.9) | 23.8 (1.5) | 48.8 (1.6) |
| KK23 w/ SimpleSE, $\tau = 0.6$ | 70.9 (1.9) | 45.4 (3.8) | 17.5 (2.9) | 40.7 (2.8) |
| KK23 w/ SimpleSE, $\tau = 0.6 \rightarrow$ KK45 w/ SimpleSE, $\tau = 0.5$ | 74.1 (1.4) | 49.9 (1.8) | 19.4 (1.4) | 43.7 (1.5) |
| $\rightarrow$ KK45 w/ SimpleSE, $\tau = 0.6$ | 76.2 (2.0) | 49.7 (1.8) | 20.6 (2.1) | **44.8** (2.0) |
| $\rightarrow$ KK45 w/ SimpleSE, $\tau = 0.7$ | 72.4 (2.0) | 48.6 (3.5) | 20.3 (2.1) | 43.2 (2.5) |
| $\rightarrow$ KK45 w/ SimpleSE, $\tau = 0.8$ | 68.4 (1.9) | 44.3 (2.0) | 18.6 (1.5) | 40.1 (1.7) |
| $\rightarrow$ KK45 w/ Oracle Verifier | 80.8 (1.2) | **60.9** (1.6) | **29.8** (2.9) | 53.3 (2.0) |
| KK23 w/ Oracle Verifier | 78.4 (1.8) | 52.6 (2.1) | 21.4 (1.4) | 46.6 (1.7) |
| KK23 w/ Oracle Verifier $\rightarrow$ KK45 w/ SimpleSE, $\tau = 0.5$ | 80.3 (1.7) | 53.7 (2.5) | 22.9 (2.4) | 48.0 (2.2) |
| $\rightarrow$ KK45 w/ SimpleSE, $\tau = 0.6$ | 77.7 (1.2) | 56.2 (1.6) | 21.6 (2.2) | 47.5 (1.7) |
| $\rightarrow$ KK45 w/ SimpleSE, $\tau = 0.7$ | 76.3 (1.5) | 53.9 (1.9) | 19.2 (1.7) | 45.4 (1.7) |
| $\rightarrow$ KK45 w/ SimpleSE, $\tau = 0.8$ | 78.7 (2.0) | 51.8 (2.2) | 19.8 (1.9) | 45.7 (2.0) |
| $\rightarrow$ KK45 w/ Oracle Verifier | **84.2** (1.6) | 60.2 (2.0) | 28.2 (1.7) | **53.3** (1.8) |

**Set-Up and Noise Reduction.** A central challenge is the noisiness of an unsupervised verifier. Smaller models in particular may mislabel solutions or produce inconsistent judgments, contaminating the preference dataset. To fairly explore self-evolution, we implement a **thresholded majority voting** method. For each candidate $\hat{y}$, the verifier is queried $n$ times, producing binary judgments $Z_j = \mathbf{1}\{\mathcal{V}^{(j)}(q, \hat{y}) = \texttt{Correct}\}$. We then compute the empirical correctness rate $\hat{p}(q, \hat{y}) = \frac{1}{n} \sum_{j=1}^{n} Z_j$. A candidate is labeled `Positive` if $\hat{p}(q, \hat{y}) \geq \tau$, `Negative` if $1 - \hat{p}(q, \hat{y}) \geq \tau$, and discarded otherwise. Note that thresholding at 0.5 falls back to regular majority voting. This procedure filters out ambiguous cases and yields high-confidence preference pairs, extracting a signal from noisy self-assessment. As shown in Figure 2, increasing the threshold effectively improves verification accuracy. This motivates our later experiments on iterative procedures.

### 3.1. Warm-Up: SimpleSE Across Model Sizes

We first examine whether SimpleSE can induce self-improvement at different model sizes. Figure 3 reports results on gemma-3-it for 1B, 4B, and 12B, with 27B included as an approximate upper bound. All models are trained on KK instances with 2–3 people and evaluated on test sets spanning 2–8 people.

At a high level, we corroborate prior work: *self-improvement can be beneficial in targeted settings* (e.g., training and testing on variations of the same task). However, self-improvement manifests differently depending on the model scale. While the 1B model shows a negligible improvement of 0.6% (7.8% $\rightarrow$ 8.4%), the 4B model achieves a 31.2% relative increase in accuracy. This suggests a non-linear scaling of self-evolution efficacy, where a minimum base accuracy (approx. 30% on KK) is required for the verifier

to provide a signal that outweighs the noise introduced during preference pair construction. Notably, the 12B model with SimpleSE approaches performance of the 27B model. Hence, using even noisy, unsupervised training data can potentially steer a model to better accuracy on certain tasks. These results motivate our following experiments that incorporate iterative feedback and curriculum learning. Our goal is to determine if *any* unsupervised approach can get close to matching the performance of oracle signals.

### 3.2. Iterative Preference Learning (Iterative SE)

We next test whether repeating the preference-learning loop closes the gap between the unsupervised SimpleSE vs. using ground-truth labels. As shown in Table 1, performance does improve across iterations, though gains diminish over time. The first iteration provides the largest boost, while subsequent ones yield smaller increments. Interestingly, training only on easier KK instances (2–3 people) improves generalization to harder ones (4–8 people): three rounds of unsupervised DPO raise accuracy from 31.0% to 44.1%, close to the 46.6% obtained with one oracle round. However, the largest gains come from adding a final oracle round (with or without SimpleSE iterations), demonstrating that a generalization gap persists even with an iterative approach.

### 3.3. Curriculum Learning (Curriculum SE)

Next, we study how scheduling problem difficulty impacts self-evolution. In *curriculum learning*, we first train on easier problems before progressing to harder ones. This contrasts with a *random mixing* baseline that uses both easy and hard problems jointly from the start. As shown in Table 2, curriculum learning consistently outperforms random mixing, for both self-improvement and the oracle setting. In the unsupervised setting, we posit that starting with simpler problems reduces verifier noise and provides more reliable supervision in early stages. Moreover, curriculum learning improves easy-to-hard transfer: training on KK with 2–3 people and then 4–5 people yields an average accuracy of 44.8%, compared to 31.0% for the base model and 41.2% for random mixing. But, we note that the same trend occurs with the oracle setting, offering little added benefit of curriculum learning for self-evolution. Overall, initial calibration is a key factor in self-improvement success.

### 3.4. The Training Cost of Self-Evolution

Finally, we analyze the computational trade-offs of the generator–verifier framework. Self-evolution requires both multiple candidate generations and multiple verifier passes. The total cost thus grows with the number of generations per query ($n_1$) and verifier passes per candidate ($n_2$). We vary $n_1$ and $n_2$ systematically across thresholds from 0.5 to 0.8 and report average performance in Figure 4.
For `gemma-3-4b-it`, threshold $\tau = 0.7$ achieves the

*Table 3*. **Results on KK for 1B, 4B, and 12B models.** Accuracy (%) is averaged over subsets with 2–3, 4–5, and 6–8 ppl., with std. dev. in parentheses. Rows compare SimpleSE at different verifier thresholds $\tau$, RevisionSE, and an oracle verifier (ForestGreen) using ground-truth labels. Across model sizes, the gap between RevisionSE and oracle supervision decreases substantially, suggesting that the RevisionSE generalization gap is strongly scale-dependent.

| Model/Algorithm | 2–3 ppl. | 4–5 ppl. | 6–8 ppl. | All |
|---|---|---|---|---|
| gemma-3-1b-it | 20.9 (1.1) | 4.9 (1.1) | 1.0 (0.4) | 7.8 (0.2) |
| SimpleSE, $\tau = 0.5$ | 15.2 (1.6) | 3.5 (0.6) | 0.8 (0.3) | 5.7 (0.6) |
| SimpleSE, $\tau = 0.6$ | 17.0 (1.8) | 2.0 (0.8) | 0.3 (0.2) | 5.6 (0.1) |
| SimpleSE, $\tau = 0.7$ | 19.0 (1.7) | 3.3 (0.8) | 0.4 (0.3) | 6.5 (0.3) |
| SimpleSE, $\tau = 0.8$ | 23.8 (2.6) | 4.5 (1.1) | 0.8 (0.4) | **8.4** (0.5) |
| Oracle Verifier | 32.6 (2.2) | 10.0 (0.9) | 0.7 (0.4) | 12.5 (0.4) |
| **RevisionSE** | 22.4 (2.4) | 4.7 (0.8) | 0.2 (0.2) | 7.8 (0.5) |
| gemma-3-4b-it | 62.0 (1.7) | 31.0 (0.9) | 10.3 (1.3) | 31.0 (0.5) |
| SimpleSE, $\tau = 0.5$ | 70.8 (1.2) | 39.1 (2.6) | 16.3 (1.7) | 38.4 (0.7) |
| SimpleSE, $\tau = 0.6$ | 70.9 (1.9) | 45.4 (3.8) | 17.5 (2.9) | 40.7 (0.9) |
| SimpleSE, $\tau = 0.7$ | 70.1 (1.6) | 43.9 (1.0) | 16.4 (1.9) | 39.6 (0.6) |
| SimpleSE, $\tau = 0.8$ | 70.4 (1.6) | 44.6 (2.1) | 16.0 (1.1) | 39.7 (0.5) |
| Oracle Verifier | 78.4 (1.8) | 52.7 (2.1) | 21.4 (1.4) | 46.6 (0.7) |
| **RevisionSE** | 75.8 (3.0) | 46.4 (2.6) | 17.1 (1.5) | **42.2** (0.4) |
| gemma-3-12b-it | 77.7 (1.7) | 51.9 (2.3) | 24.4 (1.2) | 47.5 (0.7) |
| SimpleSE, $\tau = 0.5$ | 78.3 (1.1) | 53.7 (1.5) | 24.0 (1.7) | 48.0 (0.5) |
| SimpleSE, $\tau = 0.6$ | 84.8 (1.8) | 55.0 (1.3) | 26.2 (1.3) | 51.1 (0.4) |
| SimpleSE, $\tau = 0.7$ | 83.0 (0.9) | 56.3 (0.6) | 24.0 (1.1) | 50.1 (0.2) |
| SimpleSE, $\tau = 0.8$ | 80.5 (2.1) | 53.9 (2.5) | 21.2 (2.1) | 47.5 (1.0) |
| Oracle Verifier | 86.8 (1.8) | 60.3 (1.4) | 27.0 (2.0) | 53.6 (0.5) |
| **RevisionSE** | 84.8 (1.0) | 58.7 (2.6) | **27.5** (1.1) | **52.8** (1.0) |

best balance of precision and recall, with average accuracy of 41.7%. Performance scales with $n_1$ and $n_2$, though costs grow linearly. These results highlight a practical trade-off: larger generator and verifier budgets yield higher accuracy, but moderate configurations already achieve strong results at much lower cost. As a rule of thumb, we conclude that scaling up verifier computation is typically more cost-effective than scaling up generator computation; however this may depend on the specific task and dataset.

## 4. RevisionSE: Incorporating Feedback

While single-turn verification takes advantage of some internal verification capabilities, it does not fully exploit the base model's ability to provide feedback and analyze solutions. For example, there are cases where an initial solution may be partially correct but contain errors. In these cases, the verifier model can identify these errors, going beyond just labeling the solution as incorrect. This in turn enables the generator and verifier to interact across multiple rounds. Specifically, the generator can **revise** its output in response to feedback, and it can **progressively improve the solution**.

We refer to this setup as **RevisionSE**, or *multi-turn generator–verifier verification*. RevisionSE enables iterative correction: the verifier provides feedback, and the generator revises its outputs in subsequent rounds. We ex-

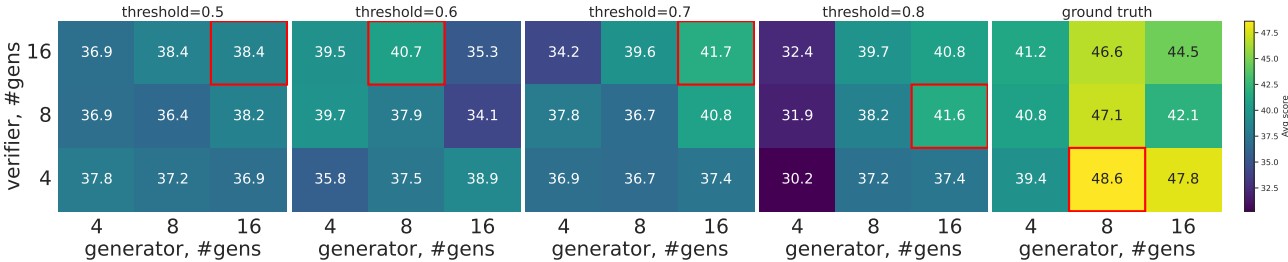

*Figure 4.* Cost–performance trade-offs in SimpleSE. Grids show average accuracy with generations $n_1$ (x-axis) and verifier passes $n_2$ (y-axis) across thresholds 0.5–0.8; the right-most plot uses oracle verification (ground-truth labels). Accuracy improves as $n_1$ and $n_2$ increase, though very high thresholds (e.g., 0.8) cause data sparsity, indicating challenges of generically applying SimpleSE.

periment with several possibilities for how to do this. The best option involves choosing the last two samples from a chain of revisions. More precisely, the RevisionSE process generates a preference pair if and only if the generator revises an incorrect solution into a correct one based on the verifier's feedback (forming a negative and positive example, respectively). We detail this in Section 2 and Figure 1.

**Results.** We evaluate RevisionSE on the KK benchmark using `gemma-3-it` (1B, 4B, and 12B) as the base model, and compare it against SimpleGV. As shown in Table 3, RevisionSE consistently outperforms SimpleGV across all thresholds and all difficulty levels. RevisionSE on `gemma-3-12b-it` achieves an average accuracy of up to 52.8%, approaching the performance of oracle ground-truth filtering (**53.6%**). This is the only case where we have seen a relatively small gap with the oracle setting, likely because the 12B model has a stronger ability to revise its own solutions, getting closer to ground truth answers. Results in Table 3 also reveal a scaling trend. For the 1B model, RevisionSE actually leads to lower accuracy than SimpleGV. This holds even for 2–3 people, where we also train on examples with 2–3 people (22.4% vs. 23.8%). On the other hand, for the 4B and 12B, we see a consistent improvement with RevisionSE, with better results compared to SimpleGV across all verifier thresholds. This demonstrates that self-improvement methods must be adaptive to model size and ability, making them harder to use in practice.

**Discussion.** Our findings from RevisionSE suggest that as model capacity grows, its dual roles as generator and verifier become increasingly effective. Intuitively, this is because the verifier feedback is more detailed, and the generator can better incorporate this feedback when revising the solution. Indeed, with the 12B model, we can nearly achieve the same performance as the oracle setting. This trend may continue or saturate with even larger models, which is an area of future work. Note that RevisionSE takes advantage of the *offline* nature of our training, where we can use natural language feedback to create better preference data.

## 5. Beyond Knights & Knaves

**Datasets.** We also consider self-improvement for other mathematical reasoning benchmarks. *GSM8K* (Cobbe et al., 2021): grade-school math word problems requiring multi-step arithmetic. *MATH500*: a medium-scale subset of the MATH benchmark (Hendrycks et al., 2021) spanning diverse levels of difficulty. *MATHHard*: the hardest subset of MATH (difficulty level 5), with advanced problem-solving. *TabMWP* (Lu et al., 2022): math word problems involving structured tabular data.

**Set-Up.** For training prompts, we use OpenThoughts3 (Guha et al., 2025) for mathematical reasoning tasks. Notably, OpenThoughts3 includes problems that are not directly verifiable (e.g., proofs and scientific question answering). This is a different setting than KK, and it explores the role of self-improvement when using a verifier that can analyze free-form outputs. To ensure a fair comparison, we include extensive prompt optimization across multiple datasets (details in Section D). We report results for the best generically applicable prompt.

**Baselines.** AZR (Zhao et al., 2025a) is an online self-play RL method that trains a model without human-curated training prompts. The model proposes code-based reasoning tasks, solves them, and receives verifiable rewards from a code executor, which is used both to validate proposed tasks and to check generated answers. Thus, AZR removes external labeled data, but still relies on an external execution environment to provide grounded rewards. INTUITOR (Zhao et al., 2025b) is an online RL method based on Reinforcement Learning from Internal Feedback. Instead of using ground-truth answers, test cases, or external reward models, it uses the model's own confidence, or self-certainty, as the reward signal for policy optimization. This makes INTUITOR closer to the closed-loop objective than methods using verifiable external rewards.

Note that these comparisons are not intended to be fully controlled baselines. Our SimpleSE experiments start from instruction-tuned models and use offline preference opti-

*Table 4.* **Results on five reasoning benchmarks.** For SimpleSE, we train with 20K samples from OpenThoughts3. We compare baselines: INTUITOR, two Absolute Zero (AZR) models, and GRPO. In the RL type column, we list whether it uses online or offline training. The supervised (Supervis.) column shows if the method uses additional labels or reward models; the environment (Environ.) column shows if it uses external tools (* from original report). Note, the open-ended nature of the questions means there is no clear oracle option.

| Model/Algorithm | RL Type | Supervision | Environment | Benchmarks | | | | |
|---|---|---|---|---|---|---|---|---|
| | | | | GSM8K | MATH500 | MATHHard | TabMWP | KK |
| *Gemma 3* | | | | | | | | |
| gemma-3-4b-it | / | / | / | **89.2*** | 75.8 (0.4) | 53.7 (0.2) | 84.5 (0.2) | 31.0 (1.3) |
| **SimpleSE** | Offline | No | No | 89.0 (0.1) | **77.4** (0.6) | **55.1** (0.4) | **87.4** (0.3) | **33.2** (0.5) |
| gemma-3-12b-it | / | / | / | 94.4* | 85.6 (0.1) | 69.1 (0.3) | 95.2 (0.2) | 47.5 (0.7) |
| *Qwen 2.5* | | | | | | | | |
| Qwen2.5-7B-Instruct | / | / | / | 90.2 (0.4) | 73.5 (0.5) | 49.7 (0.3) | 91.9 (0.2) | **18.1** (0.9) |
| Base + INTUITOR | Online | No | No | 87.3* | 75* | / | / | / |
| Base + AZR | Online | No | Yes | 84.0 (0.4) | 74.4* | 32.8 (0.5) | 68.8 (0.7) | 5.1 (0.4) |
| Base + AZR-Coder | Online | No | Yes | 83.4 (0.1) | 72.6* | 40.1 (0.7) | 78.5 (0.5) | 8.5 (0.4) |
| **SimpleSE** | Offline | No | No | **90.6** (0.1) | **76.0** (0.7) | **51.5** (0.4) | **92.3** (0.2) | 17.6 (0.5) |
| Qwen2.5-14B-Instruct | / | / | / | 94.8* | 77.1 (0.5) | 54.5 (0.3) | 93.7 (0.3) | 26.4 (0.3) |

mization, since our generator–verifier framework requires the model to follow instructions and produce reliable judgments. In contrast, AZR and INTUITOR are reported in their original online RL settings and are primarily trained from base models. We include them to contextualize the magnitude of closed-loop self-improvement relative to recent online self-evolution methods, rather than to claim that one training paradigm is intrinsically superior.

**Results.** Table 4 summarizes results on five reasoning benchmarks. For Absolute Zero (AZR) and INTUITOR, we evaluate their released models on the corresponding benchmarks, using their original report. In most cases, SimpleGV slightly improves over base models. However, compared to recent advances in reasoning with verifiable rewards or test-time scaling (Zhang et al., 2025b), this amount of self-improvement is relatively small. Moreover, we do not see the same trend as before, where SimpleGV does not approach the performance of a larger model. Section C.5 presents more experiments on scaling the number of prompts in the training data. The tapering of gains at 40k samples implies that increasing the volume of self-generated data is subject to diminishing returns.

**Remarks.** On the positive side, models can enhance their reasoning abilities through generator–verifier games. On the negative side, we see much smaller improvements than with Knights and Knaves. Considering Gemma 4B, the discrepancy between KK gains (about $+10\%$) and OpenThoughts gains (only $+1.6\%$ on MATH500, $+2.9\%$ on TabMWP) suggests that unsupervised self-evolution is sensitive to the verifiability of the training set. Unlike the deterministic KK logic, the open-ended nature of OpenThoughts prompts allows for multiple plausible but flawed reasoning paths that internal verifiers cannot reliably distinguish. Because OpenThoughts3 prompts lack automatically checkable an-

swers, we cannot definitively separate verifier error from prompt distribution shifts. The weaker self-evolution gains likely stem from the internal verifier's inability to reliably judge open-ended solution paths, though further ablation is required. Curiously, SimpleGV applied to an instruction-tuned model could outperform more intricate methods that have been implemented only for the base model (e.g., IN-TUITOR or Absolute Zero).

## 6. Discussion: Limits of Closed-Loop SE

Our work analyzes many flavors of self-evolution, providing a quantification of how well LLMs can improve their reasoning abilities without external supervision. Addressing our central question, we find that under a strict minimal formulation, SimpleSE generally leaves a persistent 8–13% gap compared to oracle supervision, with the exception of RevisionSE at the 12B scale (which reached 98.5% of oracle performance).

**Self-Evolution Works Sometimes.** We view all self-improvement as distilling a model's latent verification capabilities into a usable training signal. When the model is given the right framework to critique itself, this succeeds, and the need for ground truth solutions may diminish as model capacity increases. Our results support the 'sharpening' hypothesis (Huang et al., 2024), where self-evolution increases the model's confidence in its highest-probability paths. This is supported by our finding in Section B.3 that SimpleSE can improve Pass@1 accuracy, but Pass@32 remains largely unchanged. This indicates that closed-loop SE steers the model toward existing solution modes. SimpleSE can still outperform more sophisticated self-evolution methods like Absolute Zero or INTUITOR.

> **Takeaway 1: The Capability Threshold.**
> Closed-loop self-improvement depends on a model's internal verification skills and feedback ability. While smaller models may struggle due to internal noise and poor recall during verification, larger models (e.g., 12B+) have more ability to use multi-turn feedback and to get closer to oracle-supervised performance.

**Prerequisite Model Capability.** We do not find a clear consensus on whether larger or smaller models are more suitable for self-improvement. However, the "generalization gap" is a function of model scale, and the failure of smaller models to improve through RevisionSE (gemma-3-1b-it) highlights a capability threshold. Specifically, smaller models struggle with verifier recall and introduce noise during multi-turn feedback, frequently corrupting initially correct solutions into incorrect ones (Section B.4). Furthermore, while a 4B model can self-evolve to solve 2–3 person Knights and Knaves problems better, it struggles to provide self-evolution signals to get to a point where it can solve 6–8 person problems without ground-truth labels. This echoes work on generation–verification gaps and scaling laws (Song et al., 2024), though we note other work finds self-evolution to improve models as small as 1.5B or 3B (Zhao et al., 2025b; Zhang et al., 2025a; Huang et al., 2025b). This suggests self-improvement may be highly sensitive to the domain or model family.

> **Takeaway 2: The Generalization Limit.**
> Unsupervised self-evolution mainly sharpens a model's *confidence* in its existing solution paths rather than unlocking new reasoning capabilities. A persistent gap remains compared to post-training with oracle or human-labeled supervision. This is especially true for open-ended problems and OOD instances.

**Persistent Gap Exists for Self-Verification.** Ultimately, the limitations of SimpleSE and RevisionSE show that internal verification struggles to be a replacement for true supervision. The largest performance gains come from training with a final oracle round, demonstrating that a gap persists even with an iterative or curriculum-based approach. The gap can be seen on both easy KK instances (2–5 people) and hard instances (6–8 people). Furthermore, as observed in our OpenThoughts experiments, this gap is exacerbated in open-ended reasoning tasks where internal verifiers struggle to reliably distinguish plausible but flawed paths from truly correct ones. While internal verification can marginally improve performance on harder instances, the lack of verifiable external signals makes it difficult to improve out-of-distribution (OOD) generalization. Oracle supervision is beneficial to bootstrap models across distinct difficulty thresholds. We posit that this is because of the noise in the verifier, a direct symptom of the strict closed-loop constraint: without external code executors or test cases, the verifier is bounded by the base model's latent capabilities. Section D.1 shows that even Gemma 27B struggles to verify solutions.

## 7. Conclusion

Returning to our central question, closed-loop SE can close a meaningful fraction of the gap to oracle-supervised training, but under our strict formulation it usually does not eliminate it. Across the Knights-and-Knaves benchmark and broader downstream reasoning tasks, models improve through internal verification and feedback, with Gemma 3 4B rising from 31.0% to 44.8% on KK. However, ground-truth supervision remains stronger in most settings. The main exception is RevisionSE at larger scale: Gemma 3 12B nearly matches the oracle setting, suggesting that closed-loop SE becomes substantially more effective when the model has stronger intrinsic verification and revision capabilities. Our cost analysis further suggests a practical heuristic: scaling verifier passes is more impact-per-token than increasing candidate generation. Applying moderate confidence thresholds (e.g., $\tau = 0.7$) balances precision and recall, achieving strong self-evolution results at a lower computational cost.

**Future Work.** It would be good to explore a hybrid strategy when human-labeled data is limited: one may first use high-quality supervised training to improve the capability, then use self-evolution to calibrate or sharpen the model. Another avenue of further study is how offline self-evolution interacts with inference-time scaling methods such as self-consistency, best-of-$N$ sampling, and multi-agent verification, and explore data augmentations that increase training data diversity rather than simply scaling up self-generated data. Additionally, exploring dynamic self-evolution could offer a more compute-optimal frontier for self-improvement. Concretely, it is important to understand how the number of candidate generations and verification passes should scale based on problem complexity or model confidence.

## Impact Statement

Our work investigates the limits of self-evolution in LLMs and the extent to which models can self-improve their capabilities without oracle supervision. At a high-level, our work is about identifying when closed-loop self-verification is sufficient versus when resource-intensive oracle supervision is beneficial. Hence, our work opens up a deeper investigation to the role that specialized data should play in model training. We also find that self-improvement has limitations, and hence, any model improvements should be accompanied with rigorous testing, including comparing against models trained on human-labeled data.

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

## A. Related Work

A major trend for improving reasoning capabilities of LLMs involves inference-time techniques and a general goal of minimizing reliance on human supervision. Test-time improvement strategies provide a foundation for this, utilizing methods such as best-of-$N$ sampling, self-consistency, and majority voting to select optimal outputs from a pool of candidates (Huang et al., 2025a; Lightman et al., 2023; Zhang et al., 2025b). Recent advancements have introduced test-time training for discovery (Yuksekgonul et al., 2026) and multi-agent verification frameworks (Lifshitz et al., 2025) to further boost reasoning at inference. There is also theoretical work on the necessary prior knowledge for effective test-time scaling (Blum et al., 2026). Unlike these papers, our work focuses on a controlled empirical analysis of offline preference-based methods and a systematic evaluation of self-evolution strategies that update the underlying model parameters.

Self-evolution and reasoning capabilities have become central to modern post-training research. While standard Supervised Fine-Tuning (SFT) and Reinforcement Learning from Human Feedback (RLHF) remain the primary paradigms (Lambert et al., 2024; Gao et al., 2024; Wang et al., 2025), recent models increasingly leverage Reinforcement Learning from Verifiable Rewards (RLVR) to solve complex reasoning tasks (Comanici et al., 2025; Yang et al., 2025; DeepSeek AI, 2025). This has inspired a variety of self-evolution methods, including self-verification (Shafayat et al., 2025; Zeng et al., 2025; Piterbarg et al., 2025), self-improving VLM judges (Lin et al., 2025), and the use of internal confidence signals to impute rewards (Wen et al., 2025; Zhao et al., 2025b; Wang et al., 2024b). Some methods utilize online RL, such as R-Zero (Huang et al., 2025b), INTUITOR (Zhao et al., 2025b), LSP (Kuba et al., 2025), and EMPO (Zhang et al., 2025a). In contrast, our study provides a specific look at the offline preference learning regime, and it includes iterative feedback approaches and natural language feedback techniques (which are tailored to the offline training paradigm, and they lead to the best performance in the case of Gemma 12B).

Despite the promise of self-improvement, theoretical barriers and the risk of model collapse present significant challenges. A main concern in learning from synthetic distributions is the potential for universality-driven collapse or diminishing returns (Dey & Donoho, 2024; Shumailov et al., 2024; Wenger, 2024). Research into the generator–verifier gap (Song et al., 2024; Sun et al., 2025b; Zenil, 2026) suggests that without verifiable rewards, RL can face scaling limits (Setlur et al., 2025; Zhang et al., 2025c). Systematic evaluations indicate that the utility of synthetic data often reaches a plateau or leads to performance degradation once it exceeds a critical proportion of the total training mixture (Kang et al., 2025). Recent advancements address the performance disparity between model generation and verification by using distilled ensembles of specialized verifiers (Saad-Falcon et al., 2025). Finally, while human annotation is often viewed as the gold standard to avoid these pitfalls, it remains costly and susceptible to bias (Bassi et al., 2025; Giorgi et al., 2025; Plank, 2022).

Orthogonal to direct training are self-refinement techniques, which enable models to iteratively polish text or logical chains through self-feedback (Huang et al., 2025c; Madaan et al., 2023; Lee et al., 2025; Kim et al., 2025; Ding & Zhang, 2025). These methods can be combined with training by explicitly teaching models to self-correct (Kumar et al., 2024). Complementary research uses rubrics to evaluate model responses and moves toward a general replacement for verifiable rewards (Anugraha et al., 2025; Gunjal et al., 2025; Jayalath et al., 2025; Sharma et al., 2025). Still, a critical open question remains, regarding the extension of verifiable rewards (e.g., for math or code) to more subjective domains (Liu et al., 2025; Wu et al., 2025), alongside the challenge of maintaining high-quality synthetic prompt distributions (Yu et al., 2025).

Finally, a growing trend for better reasoning is the study of methods that minimize the need for custom-trained rewards. Going beyond RL (Zhao et al., 2025c; Ji et al., 2024), this includes reinforced self-play (Zhao et al., 2025a), synthetic code edits (Piterbarg et al., 2025), co-evolutionary collective feedback (Yuan et al., 2025), self-logits evolution (Zhang et al., 2024), and DPO extensions (Tu et al., 2025). To generate robust rewards, researchers have explored confident reasoning traces (Jang et al., 2025), external feedback models (Sun et al., 2025a), teaching reward models to "think" (Zhou et al., 2025), autorating RAG contexts (Joren et al., 2025), entropy-based methods (Zhang et al., 2025a), and ways to avoid spurious rewards (Shao et al., 2025). Our work complements these efforts, exploring the generalization gap of self-improvement compared to expert-labeled data.

## B. Additional Experiment Results

### B.1. Cross-Model Evidence on Knights-and-Knaves

To test whether the qualitative trend on KK is specific to Gemma, we additionally evaluate Qwen2.5-7B-Instruct under the same closed-loop setup. As shown in Table 5, SimpleSE and RevisionSE both improve over the base model, but still remain substantially below oracle-supervised training. This mirrors the main Gemma results and suggests that the gap to oracle

supervision is not specific to a single model family.

*Table 5.* **Qwen2.5-7B-Instruct results on KK.** Accuracy (%) is reported on easy instances with 2–3 people, harder instances with 4–8 people, and the overall average.

| Model/Algorithm | 2–3 ppl. | 4–8 ppl. | Avg. |
|---|---|---|---|
| Qwen2.5-7B-Instruct | 40.5 | 8.05 | 17.321 |
| SimpleSE | 42.375 | 10.2 | 19.393 |
| RevisionSE | 43.875 | 9.6 | 19.393 |
| Oracle Verifier | 58.625 | 19.45 | 30.643 |

## B.2. Verifier Recall as a Bottleneck

We further analyze the verifier behavior on 100 sampled KK training examples, paired with model-generated responses. The sample is balanced, with 50 correct and 50 incorrect responses. For each model, we measure whether the verifier predicts the response as correct or incorrect using threshold $\tau = 0.6$. Table 6 reports the resulting confusion matrices.

*Table 6.* **Verifier confusion matrices on 100 KK examples.** Each set contains 50 correct and 50 incorrect generated responses. Rows denote the true label and columns denote the verifier prediction. Recall is computed over truly correct responses.

| Model | Verifier | TrueP → PredP | TrueP → PredN | TrueN → PredP | TrueN → PredN |
|---|---|---|---|---|---|
| Gemma 3 4B | Base | 26 | 24 | 21 | 29 |
| Gemma 3 4B | After SimpleSE | 31 | 19 | 9 | 41 |
| Gemma 3 1B | Base | 16 | 34 | 15 | 35 |
| Gemma 3 1B | After SimpleSE | 18 | 32 | 11 | 39 |

The 4B verifier has similar initial accuracy to the 1B verifier, but much higher recall on correct responses. After SimpleSE, Gemma 3 4B improves from 55% verifier accuracy and 52% recall to 72% accuracy and 62% recall. In contrast, Gemma 3 1B improves only from 51% accuracy and 32% recall to 57% accuracy and 36% recall. This aligns with the downstream training trend: Gemma 3 4B improves substantially under SimpleSE, while Gemma 3 1B gains very little. These results suggest that verifier recall is an important bottleneck for effective self-evolution, since rejecting correct generations prevents the model from constructing useful positive preference examples.

## B.3. Pass@1 vs. Pass@k

We next test whether SE improves the model's solution coverage or primarily sharpens its single-sample behavior. Table 7 reports Pass@1 and Pass@32 for Gemma 3 4B across KK and MATH500. Across multiple rounds of SimpleSE, Pass@1 improves consistently, while Pass@32 remains nearly unchanged. This suggests that SE largely pushes the model toward solutions already reachable under sampling, rather than substantially expanding the set of problems that the model can solve with many samples.

*Table 7.* **Pass@1 vs. Pass@32 for Gemma 3 4B.** SE improves Pass@1 more than Pass@32, suggesting that it primarily sharpens existing solution modes.

| Dataset | Model/Algorithm | Pass@1 | Pass@32 |
|---|---|---|---|
| KK | gemma-3-4b-it | 31.0 | 78.0 |
| KK | SimpleSE | 40.7 | 77.4 |
| KK | SimpleSE ×2 | 43.3 | 78.2 |
| KK | SimpleSE ×3 | 44.1 | 78.8 |
| MATH500 | gemma-3-4b-it | 75.8 | 93.8 |
| MATH500 | SimpleSE | 77.4 | 94.2 |
| MATH500 | SimpleSE ×2 | 78.1 | 93.8 |
| MATH500 | SimpleSE ×3 | 78.3 | 94.0 |

## B.4. RevisionSE Step Analysis

Finally, we analyze how RevisionSE changes model outputs during multi-turn feedback. On 100 sampled KK training examples, we record the number of revision steps required for Gemma 3 4B to reach a correct answer. As shown in Table 8, successful correction typically requires 3–5 revision steps.

*Table 8.* **Number of RevisionSE steps needed to reach a correct answer for Gemma 3 4B on KK.** Most successful corrections occur after 3–5 steps.

| # Steps to become correct | $< 3$ | 3 | 4 | 5 | $> 5$ |
|---|---|---|---|---|---|
| # Samples | 4 | 29 | 42 | 16 | 9 |

We also measure how correctness changes after one revision step. Table 9 shows that RevisionSE can correct many initially wrong solutions, but also introduces non-negligible noise by turning some initially correct solutions into incorrect ones. This helps explain why revision-based self-evolution is effective only when the model is sufficiently capable: the revision process must create more useful corrections than harmful corruptions.

*Table 9.* **Correctness flips after one RevisionSE step for Gemma 3 4B on KK.** T/F denotes whether the initial or revised answer is correct.

| Flip Type | T→T | T→F | F→T | F→F |
|---|---|---|---|---|
| # Samples | 47 | 12 | 27 | 14 |

For Gemma 3 1B, the number of harmful T→F flips is close to the number of helpful F→T flips, especially on harder examples. This partially explains why RevisionSE is much less effective for the 1B model than for larger models. Overall, additional revision steps can improve solution quality, but they also introduce diminishing returns and additional noise.

# C. Experiment Details

## C.1. Training and Evaluation Settings

We consider two main training settings. **Synthetic reasoning.** Models are trained on the *Knights and Knaves* (KK) training set (restricted to instances with 2–3 people) and evaluated on the held-out KK test set covering 2–8 people. **Mathematical reasoning.** Models are trained on the *OpenThoughts3* dataset and evaluated on four benchmarks: GSM8K, MATH500, MATHHard, TabMWP, as well as the KK test set. No additional preprocessing was applied beyond the original dataset splits.

## C.2. Models and Optimization

We use instruction-tuned `gemma-3-it` models (1B, 4B, 12B) and `Qwen-2.5-7B-Instruct`. All models are fully fine-tuned (no parameter-efficient adaptation). Optimization uses AdamW with a sequence length of 4096 and batch size of 256. Training schedules are as follows:

- **Gemma-1B:** learning rate $7.5 \times 10^{-7}$, 3 epochs.

- **Gemma-4B:** learning rate $5.0 \times 10^{-7}$, 3 epochs.

- **Gemma-12B:** learning rate $2.5 \times 10^{-7}$, 3 epochs.

- **Qwen-7B:** learning rate $7.5 \times 10^{-7}$, 5 epochs.

## C.3. Generator–Verifier Setup

Unless otherwise specified, we use $n_1 = 8$ candidate generations per query and $n_2 = 16$ verifier passes per candidate. We set the confidence threshold to $\tau = 0.6$. For RevisionGV (multi-turn verification), the generator revises responses for up to 4 rounds, with the verifier providing free-form feedback that ends with a structured label.

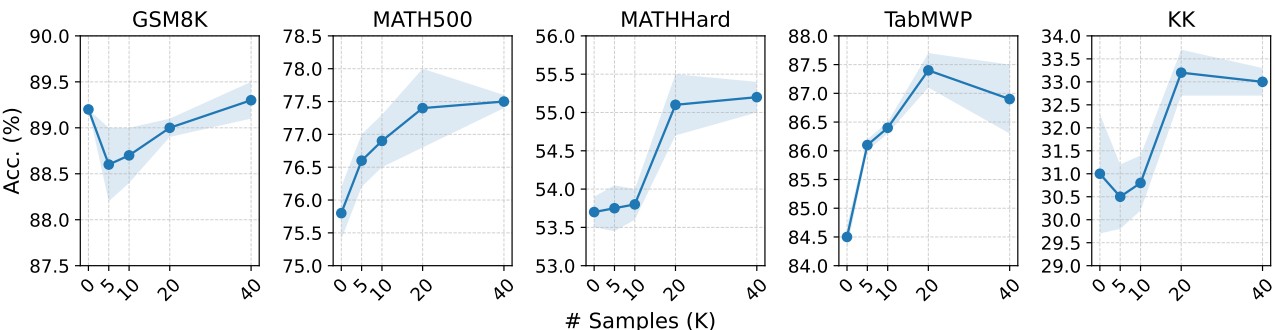

*Figure 5.* Effect of data size on SimpleSE performance. Models are trained on the OpenThoughts3 dataset. Accuracy improves across benchmarks as the number of training samples increases. For TabMWP and KK, performance slightly degrades when data increases from 20k to 40k. Subplots show mean accuracy (4 runs) with shaded standard-error regions.

### C.4. Iterative and Curriculum Learning

For iterative preference learning, we repeat the generator–verifier loop for 2–3 rounds. To isolate the effect of iteration, we reuse the same prompt set at each round rather than re-sampling. For curriculum learning, difficulty levels are determined by the KK dataset (based on the number of people). By default, models are trained on KK with 2–3 people before being evaluated on harder cases.

### C.5. Further Results on Reasoning Datasets

A natural question is how the amount of self-generated data influences downstream performance. To investigate this, we experiment with the `gemma-4b-it` model, varying the size of the preference dataset constructed from OpenThoughts3 using the generator–verifier game. We consider datasets of 5K, 10K, 20K, and 40K preference pairs (by using a comparable number of initial questions), while keeping all hyperparameters fixed.

As shown in Figure 5, enlarging the preference set yields clear gains at small–moderate scales (e.g., 5k → 20k), but improvements taper thereafter and can even regress at 40k for TabMWP and KK. This reflects *diminishing returns* from simply adding more self-generated pairs: beyond a moderate size, redundancy and verifier noise begin to dominate, suggesting that tighter filtering and greater prompt diversity are more effective than sheer volume. We note a small dip at 5k samples on GSM8K and KK; we attribute this to small-sample variance and mild prompt-distribution skew in early batches, which diminishes as the dataset grows in size and diversity.

## D. Prompts and Prompt Optimization for Verification

We provide the prompts we use for the specific KK verifier, the generic verifier, and the generic reviser. In Section D.1, we perform experiments showing how the verifier performance changes with prompts and model sizes. This complements our results in Figure 2 on verifier accuracy (for the specific KK prompt) as the model trains. For the main experiments, we use "Generic Prompt 3" for our OpenThoughts data collection.

**Knights and Knaves (KK) Specific Prompt for Verifier**

```
CRITIC_POS_SYMBOL = "CRITIC RESULT: SOLUTION IS CORRECT"
CRITIC_NEG_SYMBOL = "CRITIC RESULT: SOLUTION IS INCORRECT"

CRITIC_SYSTEM_MESSAGE = f"""You are a critic tasked with analyzing
a solution to a logical reasoning problem and determining whether
the solution correctly deduces the identities of characters
(e.g., knights or knaves). Carefully examine whether the
explanation uses valid deductive logic, correctly interprets
the statements, and exhaustively considers all cases. Pay
attention to whether contradictions are correctly identified
and if the conclusion logically follows from the analysis.

- If the solution is logically sound and complete, finalize
your critique with '{CRITIC_POS_SYMBOL}'.

- If the solution contains reasoning flaws, invalid
assumptions, missed cases, or unsupported conclusions, explain
these issues in detail and finalize with '{CRITIC_NEG_SYMBOL}'."""

CRITIC_PROMPT_TEMPLATE = (
    f"{CRITIC_SYSTEM_MESSAGE}\n\n"
    "## Problem\n{query}\n\n"
    "## Solution\n{response}\n\n"
    "## Your response"
)
```

**Generic Prompt for Verifier**

```
CRITIC_POS_SYMBOL = "CRITIC RESULT: SOLUTION IS CORRECT"
CRITIC_NEG_SYMBOL = "CRITIC RESULT: SOLUTION IS INCORRECT"

CRITIC_SYSTEM_MESSAGE = f"""You are a meticulous and critical
logic expert specializing in math, puzzles, logic and
factuality problems. Your task is to analyze a proposed
solution to the problem below and determine if it is correct.

To do this, go through the problem and then go through each
step in the answer very carefully, checking if there are any
inconsistencies or contradictions with the conditions in the
problem. End your response with '{CRITIC_POS_SYMBOL}' if the
solution has no contradictions/ inconsistencies to the
conditions in the question. Otherwise, end your response with
'{CRITIC_NEG_SYMBOL}' if the solution has one more
contradictions or inconsistencies."""

CRITIC_PROMPT_TEMPLATE = (
    f"{CRITIC_SYSTEM_MESSAGE}\n\n"
    "## Problem\n{query}\n\n"
    "## Solution\n{response}\n\n"
    "## Your response"
)
```

**Generic Revision Prompt**

```
ANALYSIS_SYMBOL = "ANALYSIS:"
REVISOR_ANSWER_SYMBOL = "REVISED SOLUTION:"

REVISOR_SYSTEM_MESSAGE = f"""You are given a logical reasoning
problem, an initial solution, and a critic's feedback on that
solution. Your task is to revise the original solution so that
it is correct, logically sound, and fully aligned with the
problem's requirements. Your revision should strictly follow
the instructions in the problem and address all the issues
raised by the critic.

Your response should be in the following format:
{ANALYSIS_SYMBOL} ...
{REVISOR_ANSWER_SYMBOL} ..."""

REVISOR_PROMPT_TEMPLATE = (
    f"{REVISOR_SYSTEM_MESSAGE}\n\n"
    "## Problem\n{query}\n\n"
    "## Solution\n{response}\n\n"
    "## Critic's Feedback\n{critic_feedback}\n\n"
    "## Your response"
)
```

## D.1. Verifier Results

We perform a deep dive here in the accuracy of the unsupervised verifier. For different datasets, we compare an unsupervised labeled (correct/incorrect) versus using a strong model (Gemini 2.5 Pro) that has access to the ground truth as the true label. We then compute the agreement with the supervised strong model as a measure of unsupervised verifier accuracy. Furthermore, we compare using three generic prompts (e.g., not specific to reasoning) against a dataset-specific prompt. We present our results in Tables 10, 11, 12, and 13.

*Table 10.* Gemma 4B, MATH, Verifier Accuracy. We sample 60 question and compare precision and accuracy (correct/incorrect labels) of an unsupervised model (no ground truth) versus supervised Gemini 2.5 Pro (with ground truth answers). For the "Single" cases we average over 3 runs, standard deviation in parens. We also evaluate taking the Majority (Maj.) over 3 samples.

| Prompt Type | Prec. Single | Acc. Single | Prec. Maj. | Acc. Maj. |
|---|---|---|---|---|
| Specific Prompt MATH | 85.1 (0.0) | 85.0 (0.0) | 85.1 | 85.0 |
| Generic Prompt 1 | **89.0 (0.1)** | **88.9 (0.8)** | **88.9** | **88.3** |
| Generic Prompt 2 | 87.0 (1.8) | 86.7 (2.7) | 87.0 | 86.7 |
| Generic Prompt 3 | 81.4 (1.4) | 80.6 (1.6) | 83.0 | 81.7 |

*Table 11.* Gemma 4B, KK, Verifier Accuracy. We sample 60 question and compare precision and accuracy (correct/incorrect labels) of an unsupervised model (no ground truth) versus supervised Gemini 2.5 Pro (with ground truth answers). For the "Single" cases we average over 3 runs, standard deviation in parens. We also evaluate taking the Majority (Maj.) over 3 samples.

| Prompt Type | Prec. Single | Acc. Single | Prec. Maj. | Acc. Maj. |
|---|---|---|---|---|
| Specific Prompt KK | 45.2 (4.1) | 53.3 (1.4) | 50.0 | 55.0 |
| Generic Prompt 1 | 52.3 (1.8) | 57.8 (2.1) | 51.3 | 56.7 |
| Generic Prompt 2 | 52.8 (2.2) | 58.3 (2.7) | 52.9 | 58.3 |
| Generic Prompt 3 | **55.1 (0.6)** | **60.0 (0.0)** | **55.2** | **60.0** |

*Table 12.* Gemma 27B, MATH, Verifier Accuracy. We sample 60 question and compare precision and accuracy (correct/incorrect labels) of an unsupervised model (no ground truth) versus supervised Gemini 2.5 Pro (with ground truth answers). For the "Single" cases we average over 3 runs, standard deviation in parens. We also evaluate taking the Majority (Maj.) over 3 samples.

| Prompt Type | Prec. Single | Acc. Single | Prec. Maj. | Acc. Maj. |
|---|---|---|---|---|
| Specific Prompt MATH | 94.0 (0.7) | 91.7 (0.0) | 94.6 | 91.7 |
| Generic Prompt 1 | 94.2 (0.8) | 94.4 (0.8) | 94.7 | 95.0 |
| Generic Prompt 2 | **92.5 (0.7)** | **92.2 (0.8)** | **91.5** | **91.7** |
| Generic Prompt 3 | 91.9 (0.8) | 90.0 (1.4) | 91.4 | 90.0 |

*Table 13.* Gemma 27B, KK, Verifier Accuracy. We sample 60 question and compare precision and accuracy (correct/incorrect labels) of an unsupervised model (no ground truth) versus supervised Gemini 2.5 Pro (with ground truth answers). For the "Single" cases we average over 3 runs, standard deviation in parens. We also evaluate taking the Majority (Maj.) over 3 samples.

| Prompt Type | Prec. Single | Acc. Single | Prec. Maj. | Acc. Maj. |
|---|---|---|---|---|
| Specific Prompt KK | 84.7 (3.2) | 68.3 (2.7) | 85.7 | 70.0 |
| Generic Prompt 1 | **90.5 (2.0)** | **90.0 (3.6)** | **91.3** | **91.7** |
| Generic Prompt 2 | 77.6 (0.8) | 75.0 (1.4) | 76.9 | 75.0 |
| Generic Prompt 3 | 80.8 (3.6) | 72.8 (4.2) | 80.0 | 73.3 |

