# OpenReview forum: "On the Generalization Gap in Self-Evolving Language Model Reasoning"
_ICML.cc/2026/Conference — ICML 2026 regular_

### Official Review · Reviewer_sKgb · 2026-03-08

**Soundness:** 3
**Presentation:** 2
**Significance:** 3
**Originality:** 2
**Overall Recommendation:** 4
**Confidence:** 3

**Summary:**

This paper presents a systematic empirical study of self-evolution for improving LLM reasoning under a strict closed-loop setting. In this setting, only an unlabeled prompt set and a single base model are available, and the model itself generates all supervision signals, including verification and feedback. The authors unify several representative self-evolution strategies within a preference optimization framework based on Direct Preference Optimization (DPO). These strategies include single-turn verification, iterative training, curriculum learning, and multi-turn revision with feedback. The unified framework allows controlled comparisons across different self-evolution mechanisms while keeping the training setup consistent. Experiments are conducted mainly on a clean and verifiable Knights and Knaves (KK) benchmark, with additional evaluations on OpenThoughts prompts and standard math benchmarks. Results show that most self-evolution strategies consistently underperform oracle-supervised training that uses ground-truth labels. An important exception is the multi-turn revision approach. When applied to Gemma-3-12B, this method significantly reduces the performance gap and approaches the oracle-supervised baseline. Overall, the paper provides an empirical characterization of the strengths and limitations of current self-evolution methods for improving reasoning in large language models.

**Compliance With Llm Reviewing Policy:**

Affirmed.

**Final Justification:**

I appreciate the authors’ response, which has largely addressed my previous concerns. I recommend to weak accept.

**Key Questions For Authors:**

1) What reference policy π_ref is used for DPO at each iteration (fixed at M0 vs updated per round)?
2) For RevisionSE, what fraction of prompts experienced an incorrect $\rightarrow$ correct flip, how many revision turns were typically required, and how does performance vary with the maximum number of revision rounds?
3) On OpenThoughts, how did you evaluate the quality of responses for problems that lack direct verifiability (spot‑checks)?
4) How robust are your findings to test‑time scaling choices (e.g., temperature, pass@k) during evaluation, and does self‑evolution interact with self‑consistency at inference?

**Limitations:**

1) Preference optimization is limited to DPO. Stronger or more robust preference methods are not tested and could change conclusions.
2) Cost–benefit comparisons across SimpleSE vs RevisionSE vs oracle rounds would be more compelling with token‑level budgets, wall‑clock.
3) The experiments focus on only a few model scales (1B, 4B, and 12B), and therefore the conclusions regarding the relationship between model scale and the effectiveness of self-evolution may still be influenced by differences in model families and architectures.
4) Although the paper highlights a persistent gap between self-evolution and oracle supervision, it does not systematically analyze whether this gap primarily arises from verifier noise, insufficient data coverage, or limitations of the optimization objective.

**Strengths And Weaknesses:**

Strengths.
1) Unifies diverse self-evolution strategies under a consistent preference-optimization lens, enabling fair, apples-to-apples comparisons. Clear, strict problem formulation (same model as generator and verifier; no external signals) isolates the generalization gap attributable to internal supervision quality.
2) Introduces practical verifier-noise control via thresholded majority voting and studies cost–accuracy trade-offs by varying generator/verifier budgets.
3) Multiple model scales (1B/4B/12B; plus Qwen-2.5-7B for OpenThoughts experiments) and several training schedules (single-round, iterative, curriculum).

Weaknesses.
1) The clarity of the presentation could be improved. The abbreviation “SE” (self-evolution) appears somewhat abruptly. In addition, some methods used in the experiments, such as INTUITOR and Absolute Zero, are included in the comparison tables without sufficient explanation of their core ideas or training setups in the main text.
2) The single‑model verifier constraint, while principled, may be overly restrictive and pessimistic; “closed‑loop” still admits options like cross‑checkpoint or multi‑verifier ensembles that are not explored.
3) The paper shows verifier accuracy vs thresholds, but omits per‑difficulty precision/recall, label‑noise rates in the final preference sets, and how these change with scale/iterations.
4) RevisionSE design choices (feedback template quality, number of turns, acceptance criteria) are under‑analyzed.

---

> ### Author Rebuttal · Authors · 2026-03-31
>
> ### W1
> Thank you for pointing this out. We will refine the text to introduce abbreviations like "SE" more smoothly and provide expanded explanations of the INTUITOR and Absolute Zero baselines in the main text.
>
> ### W2
> We agree this is a restrictive choice, but it is intentional. Our goal is not to exhaust all closed-loop variants, but to analyze several methods within a unified and clean framework, where we try to cover and compare many setups with different high-level designs
> Extensions like cross-checkpoint or multi-verifier ensembles introduce additional capacity or diversity, making it harder to attribute gains purely to self-evolution. We view these as orthogonal scaling directions rather than fundamentally different paradigms.
> Exploring them is valuable future work, but outside our scope of establishing a clean baseline under minimal assumptions.
>
>
> ### W3
>
> Per your concern, we addtionally conducted the following experiment:
>
> Using threshold = 0.6, we test gemma-3-4b 's verification performance. We randomly sampled 100 examples from the KK train set paired with gemma-3-4b as a generator’s responses, half correct and half incorrect, and checked the confusion matrix.
>
> Original:
> | |PredP|PredN|
> |:---|:---:|:---:|
> |TrueP|26|24|
> |TrueN|21|29|
>
> Acc = 55%
> Recall = 26 / 50 = 52%
>
> After SimpleSE:
> | |PredP|PredN|
> |:---|:---:|:---:|
> |TrueP|31|19|
> |TrueN|9|41|
>
> Note that self-improvement for gemma 4b is 40.7 - 31.0 = 9.7.
>
> We also tested gemma-3-1b:
>
> Original:
> | |PredP|PredN|
> |:---|:---:|:---:|
> |TrueP|16|34|
> |TrueN|15|35|
>
> Acc = 51%
> Recall = 16 / 50 = 32%
>
> After SimpleSE:
> | |PredP|PredN|
> |:---|:---:|:---:|
> |TrueP|18|32|
> |TrueN|11|39|
>
> Improvement for gemma 4b is 8.4 - 7.8 = 0.6.
>
> Observations:
> - The 4B model has similar verification acc with 1B model but much higher recall, and the 4B model self-improves more than 1B model.
> - After one round of SE, the 4B model’s recall increases more than 1B model, and there is a larger room for 4B's second-round improvement than 1B.
>
> With these evidence, we posit that **verifier's recall is a very important criterion for effective self-evolution**.
>
> ### W4
>
> We note that RevisionSE is not the primary focus of this work, but rather one working variant of self-evolution. We will add more details and analysis to better clarify its design choices and impact.
>
> ### Q1
> The reference policy is updated per round ($\mathcal{M}_{t-1}$) rather than fixed at $\mathcal{M}_0$.
>
> ### Q2
> To show some results here, we randomly sampled 100 examples from the KK training set, and we present a distribution of the number of revision steps to reach a correct answer. Here is the distribution for Gemma 3 4B. Typically, revision to correct takes 3–5 steps:
>
> | | | | | | |
> |:---|:---|:---|:---|:---|:---|
> |#steps to become correct|<3|3|4|5|>5|
> |#samples that took this many steps|4|29|42|16|9|
>
> We can also get another lens on this by measuring the before and after correctness for one revision step. Here is a table, broken down by type of flip (initial correctness -> final correctness) for Gemma 3 4B on KK:
>
>  ||T->T|T->F|F->T|F->F|
> |:---|:---|:---|:---|:---|
> |#samples|47|12|27|14|
>
> For Gemma 3 1B, the T->F is almost the same number as F->T, and by further investigation, these T->F examples are typically hard problems (# ppl>=4). This is important because it partially explains why RevisionSE is not as effective for the 1B model as it is for the larger models.
>
> Overall, we find that adding revision steps leads to a diminishing return, and sometimes added noise (e.g., for the T→F cases). We agree that doubling down on studying RevisionSE would be a valuable area of study, including having more adaptive and multi-agent critique-and-revise workflows. However, given the current breadth of our study, we leave this for future work.
>
> ### Q3
> As noted in our study, evaluating the quality of responses for problems that lack direct verifiability, such as proofs or scientific question answering, is indeed highly challenging. In our OpenThoughts experiments, we relied on the model's internal verifier using a generic evaluation prompt. However, we acknowledge that the open-ended nature of these datasets allows for plausible but flawed reasoning paths that internal verifiers cannot reliably distinguish.
>
> ### Q4
> We thank the reviewer for this insightful question. To rigorously isolate the impact of self-evolution on model parameter updates, we standardized our evaluation protocol by generating single samples at a temperature of 0.7 and averaging results across random seeds. Consequently, test-time scaling methods like self-consistency or best-of-N sampling, or multi-agent verification, are orthogonal to our core focus, which is a controlled empirical analysis of offline preference-based self-evolution. Nevertheless, we agree that understanding how offline self-evolution interacts with inference-time scaling is a promising direction for future work, and we will update our discussion to acknowledge this.

---

> > ### Author Rebuttal · Reviewer_sKgb · 2026-04-01
> >
> > The authors' rebuttal has addressed my concerns and clarified the key points. I will accordingly revise my score upward. Good Luck！

---

### Official Review · Reviewer_bhW4 · 2026-03-13

**Soundness:** 3
**Presentation:** 3
**Significance:** 3
**Originality:** 3
**Overall Recommendation:** 4
**Confidence:** 4

**Summary:**

This paper systematically evaluates the extent to which various self-evolution strategies can approximate oracle-supervised training performance. The authors impose a strict closed-loop self-evolution constraint (access only to an unlabeled prompt set and a base model) and analyze four representative methods (single-round verification, multi-turn revision, iterative training, and curriculum learning) within a unified preference optimization framework. Controlled experiments on the Knights and Knaves logical reasoning benchmark show that self-evolution yields consistent but limited gains, with a persistent performance gap of around 8–13% relative to oracle supervision. Additional validation on the OpenThoughts mathematical reasoning corpus confirms that self-evolution yields only modest improvements.

**Compliance With Llm Reviewing Policy:**

Affirmed.

**Final Justification:**

The rebuttal partially resolves my concerns. Thus, I tend to maintain the score of weak accept.

**Key Questions For Authors:**

1. What is the verifier accuracy on the OpenThoughts dataset? Can you provide an analysis similar to Figure 2 to directly validate that "poor verifiability" is the primary driver of reduced self-evolution gains?
2. Is the comparison between SimpleSE, INTUITOR, and AZR in Table 4 fair? Under controlled base model type and external resource access, would SimpleSE still maintain its advantage?

**Limitations:**

The analysis is primarily confined to offline preference learning under the DPO framework, without covering online RL methods based on policy gradients (e.g., PPO, GRPO) under the same closed-loop constraints.
Additionally, the experimental scale is limited to 12B; whether the conclusions apply to the self-evolution limits of much larger models (100B+) remains unclear.

**Strengths And Weaknesses:**

## Strengths

1. The paper's core contribution lies in bringing a range of existing methods into a unified framework for controlled comparison, rather than proposing new algorithms.
2. The experimental coverage is thorough across model scale (1B, 4B, 12B), strategy complexity (SimpleSE, RevisionSE, iterative training, curriculum learning), and computational cost analysis (Figure 4).
3. The paper's core findings have important community guidance value: self-evolution under closed-loop conditions exhibits a persistent gap, and gains are disproportionate to computational cost. This conclusion provides important counter-calibration to the large volume of recent work claiming substantial improvements from self-evolution.

## Weakness

1. The paper attributes the generalization gap between KK and OpenThoughts to "poor verifiability of open-ended questions," but this explanation is not directly validated: no quantitative analysis of verifier accuracy on OpenThoughts subsets is provided (analogous to Figure 2 for KK), making it difficult to disentangle "task verifiability" from "prompt distribution differences."
2. Table 4 directly compares SimpleSE against online RL methods such as INTUITOR and Absolute Zero, but these methods differ fundamentally in base model type (base vs. instruction-tuned) and external resource access. The paper's explanation of this asymmetry is insufficient, potentially misleading readers in interpreting SimpleSE's apparent advantage.

---

> ### Author Rebuttal · Authors · 2026-03-31
>
> Thank you for the constructive and generally positive review! We are glad that you found our core contributions compelling, including our unified framework and controlled comparisons. Also, we appreciate that you think our experiment are thorough and our findings have the potential for impacting the larger community.
>
> We aim to address your concerns below, and we are happy to answer further questions.
>
> ### W1 & Q1
>
> We agree that we cannot reliably compute verifier accuracy on OpenThoughts to prove causal attribution. To fix this, we will Section 5 to read: "Because OpenThoughts problems lack automatically checkable answers, we cannot definitively separate verifier error from prompt distribution shifts. The weaker self-evolution gains likely stem from the internal verifier's inability to reliably judge open-ended solution paths, though further ablation is required."
>
> By “poor verifiability,” we mean that many OpenThoughts problems (e.g., proofs, coding, scientific QA) lack deterministic or automatically checkable answers. As a result, unlike KK, we cannot reliably compute verifier accuracy or construct a confusion matrix. This inherent ambiguity makes it difficult to disentangle verifier errors from multiple plausible solution paths, which we believe is the primary source of weaker self-evolution gains. As mentioned above, we will clarify this limitation and avoid over-claiming.
>
> ### W2 & Q2
>
> We agree that Table 4 is not a strictly controlled comparison. Our intent was NOT to claim that SimpleSE is inherently superior to methods like INTUITOR or Absolute Zero, but to **contextualize performance**: a simple offline method on an instruction-tuned model versus more complex, compute-intensive methods on base models.
>
> A fully controlled comparison is challenging in both directions. Applying SimpleSE to raw base models would likely underperform, as these models lack the instruction-following and verification capabilities needed to bootstrap self-evolution. Conversely, applying RL methods on instruction-tuned models would introduce additional confounders. We will revise the text to make this asymmetry explicit and prevent misinterpretation.
>
> ### Additional Clarification
>
> We agree that evaluating online RL methods under the same closed-loop constraints, as well as scaling to larger models (e.g., 100B+), are important future directions. While our experiments are limited to 12B due to compute constraints, the strong performance of RevisionSE at this scale suggests that improved intrinsic verification ability at larger scales could further narrow the gap.

---

### Official Review · Reviewer_zMoa · 2026-03-14

**Soundness:** 2
**Presentation:** 3
**Significance:** 2
**Originality:** 2
**Overall Recommendation:** 4
**Confidence:** 3

**Summary:**

The paper studies self-evolution of LLM in which a model may improve only from unlabeled prompts plus its own internally generated signals, with no external labels or verifiable environment. It considers several such strategies inside a preference-learning setup and evaluates them primarily on Knights and Knaves benchmark. The central empirical claim is that self-evolution usually helps relative to the base model, but remains meaningfully worse than oracle-labeled preference optimization. Based on these observations the authors suggest a multi-turn self-evolution strategy before supervised training.

**Compliance With Llm Reviewing Policy:**

Affirmed.

**Final Justification:**

The rebuttal addressed my main concerns and changed my evaluation.

**Key Questions For Authors:**

- Is self-evolution only boosting model's confidence, i.e. pushing model's pass-1 accuracy towards pass-$\infty$, or learning some new knowledge outside the base distribution?
- How should the formal verifier be defined so that it can cover both binary judgment methods and RevisionSE-style free-form critique?
- What cost definition underlies the statement that verifier scaling is more cost-effective than generator scaling? Could you add a token-normalized or FLOP-normalized analysis to support that conclusion?

**Limitations:**

see above

**Strengths And Weaknesses:**

Strengths:
- The paper asks a clean and important question of how far a model can improve using only unlabeled prompts and its own internal signals. The experimental design is well structured with Knights-and-Knaves giving a controllable setting where difficulty can be manipulated, verifier quality can be measured, and easy-to-hard curriculum effects can be studied in a fairly interpretable way.
- The central empirical message is interesting and potentially useful. The larger-model results, especially the near-oracle behavior of the $12$B revision-based setup on Knights-and-Knaves, are intriguing and suggest a real scaling trend worth following up.

Weaknesses
- The paper studies a narrow class of tasks and LLM models, so the significance of the conclusions is limited.
- All observations appear to be quite expected and not surprising, there lacks a deeper insight. However, I'd be happy to increase my rating if these two weaknesses are addressed in revision.

---

> ### Author Rebuttal · Authors · 2026-03-31
>
> Thank you for your constructive feedback and for noting the merits of our experimental design. We address your comments and concerns below.
>
> ### W1
> We conducted most of our experiments on the Knights and Knaves benchmark intentionally because it provides deterministic solutions, mitigates data contamination concerns, and offers a clear difficulty hierarchy. In Section 5, we also supplemented this with the OpenThoughts corpus for real-world downstream reasoning problems to test self-evolution on data with open-ended verifiability. While we considered deviating further from reasoning tasks (e.g., studying alignment or instruction following), we were worried that the noise in evaluating and verifying more subjective tasks may introduce extra confounding factors and make the conclusions less reliable.
>
> We have considered multiple model scales (1B, 4B, 12B) and two model families (Qwen and Gemma). Per your concern, we hereby supplement Qwen’s result on KK to show more results on different models:
>
> |Model|2-3ppl|4-8ppl|Avg|
> |:---|:---|:---|:---|
> |Qwen2.5-7B-Instruct|40.5|8.05|17.321|
> |SimpleSE|42.375|10.2|19.393|
> |RevisionSE|43.875|9.6|19.393|
> |Oracle|58.625|19.45|30.643|
>
> ### W2 & Q1
>
> Thank you so much for raising concerns about limited insights. First, we've already provided non-trivial insights, surfacing many of nuances in the paper by comparing several flavors of self-evolution techniques, including iterative and curriculum approaches. Here, we will add some additional findings to the paper based on the reviews so far, which we describe below.
>
> Insight 1: **Verification recall matters for successful self-evolution**
>
> Using threshold = 0.6, we test gemma-3-4b 's verification performance. We randomly sampled 100 examples from the KK train set paired with gemma-3-4b as a generator’s responses, half correct and half incorrect, and checked the confusion matrix.
>
> Original:
> | |PredP|PredN|
> |:---|:---:|:---:|
> |TrueP|26|24|
> |TrueN|21|29|
>
> Acc = 55%
> Recall = 26 / 50 = 52%
>
> After SimpleSE:
> | |PredP|PredN|
> |:---|:---:|:---:|
> |TrueP|31|19|
> |TrueN|9|41|
>
> Note that self-improvement for gemma 4b is 40.7 - 31.0 = 9.7.
>
> We also tested gemma-3-1b:
>
> Original:
> | |PredP|PredN|
> |:---|:---:|:---:|
> |TrueP|16|34|
> |TrueN|15|35|
>
> Acc = 51%
> Recall = 16 / 50 = 32%
>
> After SimpleSE:
> | |PredP|PredN|
> |:---|:---:|:---:|
> |TrueP|18|32|
> |TrueN|11|39|
>
> Improvement for gemma 4b is 8.4 - 7.8 = 0.6.
>
> Observations:
> - The 4B model has similar verification acc with 1B model but much higher recall, and the 4B model self-improves more than 1B model.
> - After one round of SE, the 4B model’s recall increases more than 1B model, and there is a larger room for 4B's second-round improvement than 1B.
>
> With these evidence, we posit that verifier's recall is a very important criterion for effective self-evolution.
>
> Insight 2: **SE pushes pass@1 towards pass@k, but does not necessarily improve pass@k**
>
> Our current evaluation specifically tests easy-to-hard generalization by training on simple instances (2-3 people) and evaluating on harder ones (4-8 people). The fact that *self-evolution allows models to solve harder examples than they encountered during training* already suggests it is doing more than just boosting pass-1 accuracy.
>
> On the other hand, SE does seem to increase model confidence and consistency in its answers, where pass@1 goes up significantly, but pass@k only improves a little. Results (again with the subsampled 100 problems):
>
> On KK:
> |Model|pass@1|pass@32|
> |:---|:---:|:---:|
> |gemma-3-4b-it|31.0|78.0|
> |SimpleSE|40.7|77.4|
> |SimpleSEx2|43.3|78.2|
> |SimpleSEx3|44.1|78.8|
>
>
> On MATH500:
> |Model|pass@1|pass@32|
> |:---|:---:|:---:|
> |gemma-3-4b-it|75.8|93.8|
> |SimpleSE|77.4|94.2|
> |SimpleSEx2|78.1|93.8|
> |SimpleSEx3|78.3|94.0|
>
>
> After multiple rounds of SE, pass@1 increases but pass@32 fluctuates within a range without obvious improvements.
>
> ### Q2
>
> We model **RevisionSE** as a multi-turn interaction ($T>1$) between a generator $\mathcal{G}$ and verifier $\mathcal{V}$, where natural language feedback refines incorrect drafts.
>
> 1. **Critique**: Given query $q$ and draft $\hat{y}^{(t)}$, the verifier produces feedback:
>    $$c^{(t)} \sim \mathcal{V}(\cdot \mid q, \hat{y}^{(t)})$$
>
> 2. **Revision**: The generator updates the draft using the critique:
>    $$\hat{y}^{(t+1)} \sim \mathcal{G}(\cdot \mid q, \text{Prompt}_{rev}(\hat{y}^{(t)}, c^{(t)}))$$
>
> 3. **Preference Pair**: If revision corrects the error, construct:
>    $$y_l = \hat{y}^{(t)}, \quad y_w = \hat{y}^{(t+1)}$$
>    $$\text{s.t. } \mathcal{V}(q,\hat{y}^{(t)})=\text{Incorrect},\ \mathcal{V}(q,\hat{y}^{(t+1)})=\text{Correct}$$
>
> ### Q3
> Our analysis in Figure 4 tracks computational cost via the number of candidate generations and verifier passes. We will update this section to include token-normalized estimates to make the cost-accuracy trade-offs more explicit.

---

> > ### Author Rebuttal · Reviewer_zMoa · 2026-04-07
> >
> > Thank you for the response. My questions are addressed, so I'd like to raise my score.

---

### Official Review · Reviewer_5piB · 2026-03-16

**Soundness:** 2
**Presentation:** 3
**Significance:** 3
**Originality:** 3
**Overall Recommendation:** 4
**Confidence:** 5

**Summary:**

The paper studies the recently popular topic of self evolution / self improvement (during training) in LLMs. They compare the benefits and cel of these methods compared to cases with oracle supervision. They find that although these methods improve and generalize to harder problems, especially when the learning happens via a curriculum, but only upto a certain limit which currently is less than oracle supervision.

**Compliance With Llm Reviewing Policy:**

Affirmed.

**Final Justification:**

Thanks for the author response and the additional results. I am okay with the study being scoped. But, I think that this should be reflected in the title of the paper. I would recommend that authors make the title more clear to reflect the contributions of the paper. I keep my score.

**Key Questions For Authors:**

Please see weaknesses

**Limitations:**

Yes

**Strengths And Weaknesses:**

Strengths:
1. The paper is timely and address the important research topic of self-evolution and it's comparison with oracle supervision
2. The paper is well written
3. The conclusions provide good insights for the community.

Weaknesses:
1. The paper analyses a limited set of self-evolution techniques, and most of these techniques are known to saturate performance. The paper does not explore the generator-verifier adversarial game. Let's say you want to improve on the ability X, then the game is formulated as a competition between a generator and a verifier (possibly generative verifier)  where the generator generates some difficult to solve task for the verifier and the verifier tries to solve it. For both, the reward is whether they are able to deceive the other party or not (in addition to some rule based task specific rewards). For instance the work "Adversarial Training for Process Reward Models" by Juneja et.al. It is not clear what is the ceiling of these methods.

2. The paper only considers reasoning tasks, whereas self-improvement might have different effect on different types of tasks. I would suggest increasing the horizon of tasks.

3. The method also does not compare with darwin-godel machines for self evolution.

4. In such kind of paper, I believe, only empirical validation is not enough. I would strongly recommend adding some qualitative studies as to exactly why these models are not able to improve. What are the kind of solutions they explore, what mistakes they make repeatedly that cannot be improved via self evolution. This would provide the community more insights into how to develop methods for self evolution in a principled way.

---

> ### Author Rebuttal · Authors · 2026-03-31
>
> Thanks for the constructive & positive review.
>
> ### W1
> Thank you for raising this question. We focus on a controlled empirical analysis of offline preference-learning-based methods under a strict closed-loop constraint. We acknowledge the value of alternate setups such as the adversarial games, and indeed it is possible that other techniques could lead to better self-evolution performance. However, our goal was to systematically analyze methods within a unified and clean framework, where we try to cover and compare many setups with different high-level designs, and this kind of systematic study has not been done before. The work you mentioned involves training process reward models, but deviates from our setup in some ways, such as optimizing the model with an external PRM rather than self-evolving the model. Our current scope strictly focuses on self-evolving the model weights via preference optimization without external PRMs, to establish a minimal baseline. However, we agree that combining our closed-loop setup with an adversarial PRM training game is a good next step.
>
> ### W2
> We conducted most of our experiments on the Knights and Knaves benchmark **intentionally** because it provides deterministic solutions, mitigates data contamination concerns, and offers a clear difficulty hierarchy. In Section 5, we also supplemented this with the OpenThoughts corpus for real-world downstream reasoning problems to test self-evolution on data open-ended verifiability. While we considered deviating further from reasoning tasks (e.g., studying alignment or instruction following), we were concerned that the noise in evaluating and verifying more subjective tasks may introduce extra confounding factors and make the conclusions less reliable.
>
> ### W3
> While Darwin-Gödel machines achieve self-improvement through symbolic, verified source-code rewriting, our research empirically analyzes gradient-based post-training. We focus on updating model weights via preference optimization and natural language feedback. Because these paradigms operate under fundamentally different mechanisms (symbolic code-rewriting versus continuous-space weight updates), a direct empirical comparison falls outside the scope of our current study.
>
> ### W4
> Qualitative studies would provide deeper insights into model failure modes. We will add an appendix section detailing specific examples of verifier noise and exactly why these models fail to self-improve. Here we show two abbreviated examples of failed verification from KK training set (4-5 ppl.):
>
> #### Case 1: True generation, but verified as False (Valid reasoning rejected)
> - Q: [...] You meet 4 inhabitants: Aiden, Mason, Zoey, and Alexander. [...] So who is a knight and who is a knave?
> - A: [...] The situation is consistent: Aiden is a knave, Alexander is a knight, Mason is a knight, and Zoey is a knight. [...]  (1) Aiden is a knave. (2) Alexander is a knight. (3) Mason is a knight. (4) Zoey is a knight.
> - CRITIC RATIONALE: Let's analyze this puzzle carefully. [...] Let's suppose Zoey is a Knave. Then Aiden is a Knave and Mason is a Knave. [...] Conclusion: (1) Aiden is a knave. (2) Alexander is a knight. (3) Mason is a knave. (4) Zoey is a knave.
> - CRITIC RESULT: SOLUTION IS WRONG.
> - Insight: The verifier fails because generating a verification rationale for complex logic is just as difficult as generating the solution itself. Instead of rigorously checking the generator's valid proof, the critic attempts its own independent chain-of-thought, gets tangled in the complex nested conditionals like falsely deducing "Mason is a knave", and incorrectly penalizes a valid generation based on its own hallucinated logic.
>
> #### Case 2: False generation, but verified as True (Flawed reasoning accepted)
> - Q: [...] You meet 5 inhabitants: Aurora, Jack, Joseph, Emma, and Oliver. [...] So who is a knight and who is a knave?
> - A: [...] 8. Determine Emma's Status [...] Emma is telling the truth, so Emma is a knight. 9. Therefore, Jack must be a knave. [...] (1) Aurora is a knave. (2) Jack is a knave. (3) Joseph is a knave. (4) Emma is a knight. (5) Oliver is a knight.
> - CRITIC RATIONALE: Let's analyze this problem systematically. [...] 7. Since Emma is a knave [...] Emma’s statement must be false, aligning with her being a knave. 8. Now, let's reconsider Jack’s statement [...] Therefore, Jack is a knight. CONCLUSION: (1) Aurora is a knave. (2) Jack is a knight. (3) Joseph is a knave. (4) Emma is a knave. (5) Oliver is a knight.
> - CRITIC RESULT: SOLUTION IS CORRECT.
> - Insight: The verifier fails because it lacks the capacity for global consistency checking. It acts as a superficial proofreader; as long as the local, step-by-step deductions sound highly plausible, it rubber-stamps the output as "correct" (remarkably, even while its own rationale arrives at a completely different conclusion than the generator's answer). The critic fundamentally lacks a reliable grounding mechanism to catch internal contradictions.

---

> > ### Author Rebuttal · Reviewer_5piB · 2026-04-04
> >
> > The responses for W1,2,3 make it look like the study is **too narrowly scoped** and **does NOT give the correct view** of whether the current methods for self evolution work or not. If we are to make a negative claim (on an entire direction of research) then we should be showing it on multiple benchmarks and methods. The bar for these papers is generally high, as also described in the NeurIPS Main Track Handbook (https://neurips.cc/Conferences/2026/MainTrackHandbook):
> >
> > "Negative Results: The main contribution is in understanding a negative result. The bar for these submissions is expected to be high."
> >
> > My initial assessment was positive because understanding self-evolution is very important for the advancement of the field. **BUT** after the rebuttal, it seems like the **scope of this study is quite limited to make a big negative claim.** Still, I will not decrease my score, but if the authors are able to give a reason why the study is comprehensive, then I am willing to raise my score.

---

> > > ### Author Response · Authors · 2026-04-07
> > >
> > > Thank you for the thoughtful follow-up.
> > >
> > > We would like to clarify that our paper is **not making a claim about all forms of self-evolution**, but instead studies a specific and widely used formulation. We believe the paper is comprehensive **for the research question it asks**. We do not study
> > >
> > > RQ1: *“what is the ultimate ceiling of every self-evolution method,”*
> > >
> > > but rather
> > >
> > > RQ2: *“how far can self-evolution go under a clean, closed-loop setup, across representative offline strategies, model scales, and both controlled and real-world reasoning regimes?”*
> > >
> > > Our claim is not that all forms of self-evolution reach a limit. Instead, we focus on a strict closed-loop setting where the model has access only to unlabeled prompts and internally generated supervision, a setting that **many existing methods implicitly rely on**. Under this formulation, we consistently observe a gap to oracle supervision. We view this as a **controlled empirical baseline**, not a statement about the entire research direction.
> > >
> > > ---
> > >
> > > ## Recap
> > >
> > > Within this RQ2 formulation, we believe the study is reasonably broad:
> > >
> > > - **Representative methods.** We study four qualitatively different strategies in one unified framework, and also compare to popular online methods (Table 4), so the **conclusions are not tied to a single approach**.
> > > - **Controlled + real-world regimes.** We intentionally use KK because it is deterministic, verifiable, resistant to contamination concerns, and has a built-in difficulty hierarchy for studying easy-to-hard generalization, and extend to OpenThoughts with evaluation on standard reasoning benchmarks, so **results are not tied to a single dataset**.
> > > - **Model scale and cross-model evidence.** On KK, Gemma 4B improves from 31.0 to 44.8 across methods vs. 53.3 oracle, while Gemma 12B RevisionSE reaches 52.8 vs. 53.6 oracle, showing that **self-evolution can become effective when intrinsic verification is strong enough**.
> > >
> > > ---
> > >
> > > ## New Results and Changes
> > >
> > > We agree the scope should be clearer and will revise the paper accordingly to explicitly frame the claim as RQ2 (not RQ1). To further address breadth, we additionally include a factuality task:
> > >
> > > ### Beyond Reasoning
> > >
> > > We evaluate on MuSiQue, a widely used factuality benchmark for multi-hop question answering over context. We conduct SE on the full training dataset (~19k) and evaluate on a 1k test subset to control cost (following prior work’s setting [1,2]):
> > >
> > > | Model | Avg F1 |
> > > |---------------------|--------|
> > > | Qwen2.5-7B-Instruct | 7.1 |
> > > | SimpleSE | 7.3 |
> > > | SimpleSE x2 | 7.4 |
> > > | RevisionSE | 7.4 |
> > > | Oracle | 9.2 |
> > >
> > > We observe the same qualitative trend: SE yields consistent but modest gains, with a non-trivial gap to oracle supervision. This suggests the phenomenon is **not limited to reasoning tasks**.
> > >
> > > ### Additions
> > >
> > > We also strengthened the paper with: (please see response to Reviewer zMoa)
> > > - cross-model evidence (Qwen showing similar trends),
> > > - diagnostics identifying verifier recall as a bottleneck,
> > > - analysis showing gains in pass@1 more than pass@k
> > >
> > > These additions emphasize that the paper is not only a negative result, but also identifies **when self-evolution works and what limits it**.
> > >
> > > ---
> > >
> > > ## Concluding thoughts
> > >
> > > We hope this clarification makes clear that the paper provides a **scoped but systematic empirical characterization of closed-loop self-evolution**, rather than a blanket claim about the entire direction. We would appreciate your reconsideration in that light.
> > >
> > > ---
> > >
> > > [1] Gutiérrez, Bernal J., et al. "Hipporag: Neurobiologically inspired long-term memory for large language models." Advances in neural information processing systems 37 (2024): 59532-59569.
> > >
> > > [2] Zhang, Nan, et al. "When reasoning meets compression: Understanding the effects of llms compression on large reasoning models." arXiv preprint arXiv:2504.02010 (2025).

---

### Decision · Program_Chairs · 2026-04-30

**Decision:**

Accept (regular)

**Comment:**

This paper studies the empirical limits of LLM self-evolution under a strict, closed-loop setup using offline preference optimization. The reviewers find that the paper is timely and well-written, and asks a clean, important research question.

Concerns have been raised on the narrow scope of evaluated tasks and methods (Reviewers 5piB, zMoa), the lack of deeper qualitative insights into verifier failure modes (Reviewers 5piB, zMoa), the difficulty of disentangling task verifiability from distribution shifts on the OpenThoughts dataset (Reviewer bhW4), unfair comparisons to online RL baselines (Reviewer bhW4), and the omission of granular metrics like verifier recall and RevisionSE step-analysis (Reviewer sKgb).

During rebuttal, the authors clarified their focus, included additional experiments, and revised the texts to frame the paper's limitations. While the scope and depth of analysis remain somewhat limited, I judge that the paper's clear experimental design and consistent findings provide sufficient evidence to support its central claim. Therefore, a weak accept is recommended.